

# Identification and classification of urban micro-vulnerabilities in tsunami evacuation routes for the city of Iquique, Chile

Gonzalo Álvarez[1,3], Marco Quiroz[1,3], Jorge León[2,3], and Rodrigo Cienfuegos[1,3]

[1]Departamento de Ingeniería Hidráulica y Ambiental, Pontificia Universidad Católica de Chile, Santiago 7820436, Chile
[2]Departamento de Arquitectura, Universidad Técnica Federico Santa María, Valparaíso 2390123, Chile
[3]Centro de Investigación para la Gestión Integrada del Riesgo de Desastres (CIGIDEN), CONICYT/FONDAP/15110017, Santiago 7820436, Chile

*Correspondence to:* Rodrigo Cienfuegos (racienfu@ing.puc.cl)

**Abstract.** Many coastal cities around the world are threatened by tsunamis; some of these events have caused great impacts in recent times. The loss of human lives is the main cause of concern of the authorities, and evacuation planning has been recognized as one of the best tools for safeguarding the population. In this context, urban design appears to be critical for the execution of prompt and efficient evacuation processes to safe areas; however, evacuation assessment has been traditionally

carried out at a large urban scale, mostly taking into consideration urban morphology and connectivity. In the present work, urban spaces available for tsunami evacuation are explored in detail by developing a methodology to identify and classify urban micro-vulnerabilities that may reduce the capacity of the evacuation routes and hinder evacuees' safety. The method is applied to the Chilean city of Iquique, affected by an earthquake and subsequent tsunami in 2014.

## 1   Introduction

Most of the world's commercial activity takes place in port cities, which are important areas of resources exchange, tourism and recreation. Coastal cities have undergone major urban and population growth in recent decades, a trend that is expected to continue in the future (Neumann et al., 2015). Coastal growth increases the exposure of people and property to natural hazards and thus has a negative consequence on the risk level on urban settlements (Jongman et al., 2012; Kron, 2013).

Tsunamis are one of the most challenging coastal threats for communities located in subduction zones where the rapid arrival
of tsunami waves can produce significant damage and loss of lives once they reach the coast. Tsunami risk is determined by combining the probability of being reached by these waves, the degree of exposure, and the vulnerability of the population and the physical infrastructure. Mitigation of tsunami risks in coastal cities requires the evaluation of each of these dimensions (hazard, exposure, and vulnerability) (Kron, 2013), while improving their resilience requires additionally to evaluate their capacity to absorb disturbances, reorganize and adapt, keeping its principal functions in operation (Birkmann, 2006; Cutter
et al., 2008).

Recent experiences and a better understanding of the tsunami hydrodynamics have enhanced our ability to mitigate the consequences derived from this natural hazard. This knowledge has led to the design of countermeasures, which may be cataloged as structural and non-structural. The former are long-term countermeasures such as relocation to elevated zones and

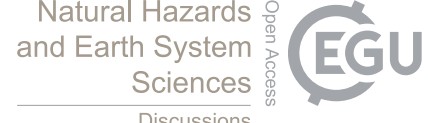



the construction of large civil engineering defenses (i.e. sea walls, breakwaters, or sea gates) which are conceived to reduce the energy of tsunami waves prior to reaching populated areas. Among the non-structural countermeasures, early warning systems, evacuation planning, and rapid assistance in response to disasters, are considered a priority to reduce the potential impacts of tsunamis. In addition, other countermeasures have been recognized, such as urban planning, building codes, and recovery plans

(Bernard and Robinson, 2009; Shuto and Fujima, 2009).

The 2011 Great East Japan Tsunami showed that structural countermeasures can be insufficient, due to design and historical data limitations, as the magnitude of the tsunamigenic earthquake and the height of the tsunami can be greater than expected. However, such countermeasures can help in reducing the impact of the tsunami waves in terms of their heights and arrival times at the coast Mori et al. (2013); Fraser et al. (2013). Global experiences have demonstrated that the most effective method

of saving lives against tsunamis is a prompt evacuation (Shuto, 2005; Suppasri et al., 2013). Therefore, the implementation of warning systems and evacuation preparedness are very important as the combination of different types of countermeasures is crucial in reducing the losses caused by large-magnitude tsunamis (Suppasri et al., 2013). In this regard, it has been recognized that it is important to promote constant investment in education, planning, monitoring and improvements of the urban infrastructure (Scheer et al., 2012; Esteban et al., 2013). This is especially important in developing countries, where the use of large

structural countermeasures is uncommon due to their high costs and slow implementation times.

Essential information for a better preparedness in case of future disasters can be obtained from post-tsunami surveys where maximum runup levels and inland inundation are determined (Kong, 2011). Chile is among the countries with the longest history of tsunamigenic earthquakes and in 2010 it faced the most deadly tsunami in Latin America in recent decades, with a death toll of 156 and 25 missing people (Huerta, 2011) and maximum runup that reached 29 m in the city of Constitución (Fritz

et al., 2011). On this basis, in the case of the 2014 and 2015 tsunamis, the maximum runup was 4.63 m in Caleta Camarones, with no deaths (Catalán et al., 2015), and 13.6 m in La Cebada, with 15 fatalities (Contreras-López et al., 2016) respectively.

Some structural countermeasures were implemented in the most affected zones in the aftermath of this event, specifically in the coastal area of the Bío-Bío Region, including promenades, low sea walls and, to a greater extent, relocation of populated areas to higher elevations (Khew et al., 2015). However, these measures are not necessarily representative of the tsunami

mitigation strategy adopted at the national level. In the Chilean case, most of the efforts have been aimed at fostering tsunami risk awareness through education, evacuation drills, and installation of signage along escape routes.

The present study aims at developing a sound methodology to identify and classify potential *micro-vulnerabilities* in the urban space that may difficult pedestrian evacuation processes. Specifically, the analysis is focused on the physical aspects of the built environment that can contribute to urban vulnerability, in particular in the case of tsunamis and their related evacuation

processes. The criteria used for the identification of these aspects are based on the literature examining how the physical characteristics of indoor and outdoor spaces contribute to the safety of people against tsunamis and other hazards (Ciborowski, 1982; Preuss et al., 2001; Ercolano, 2008; He and Xu, 2012). The final objective of this research is to contribute to better design and use of public spaces, particularly evacuation routes, in order to promote the resilience of tsunami-prone coastal cities. This topic will be analyzed in the city of Iquique, located in northern Chile, which is susceptible to large-magnitude earthquakes

and their ensuing tsunamis, and currently exhibits significant urban vulnerability problems for evacuation (Walker, 2013).



This article is structured in the following manner. First, a theoretical background is provided about the role of appropriate urban forms in mitigating tsunami-related vulnerability. Second, the study area in Iquique, alongside with its coping strategies to mitigate the tsunami hazard, are described. Third, the method for quantifying micro-vulnerabilities is detailed in order to subsequently present the obtained results. Finally, the possible consequences of the existence of micro-vulnerabilities on

evacuation routes and the measures proposed to decrease tsunami risk in coastal cities are discussed.

## 2   Gaps in tsunami evacuation planning: Linking macro and micro urban scales

Post-disaster reconstruction is an opportunity to use urban design in view of obtaining a more resilient city without neglecting economic development and the quality of life of the affected populations (Kennedy et al., 2008; Liu et al., 2014; Yi and Yang, 2014). The need to implement measures in a short time in order to guarantee the wellbeing of those affected is the source

of a large part of the measures and studies that focus on reconstruction and the development of post-disaster recovery plans (Siembieda et al., 2012; Steinberg, 2007; Spaling and Vroom, 2007; Johnson et al., 2006; Platt et al., 2002). However, only a few of these address the rethinking of the cities' urban design (Ishikawa, 2002; Liu et al., 2014; León and March, 2014, 2016; Ciborowski, 1982).

Ishikawa (2002) describes the historical evolution of landscape planning aimed at mitigating the risk of urban fires (as the

result of large earthquakes); she underlines the role of open spaces (e.g. parks and streets) and the improvement of river banks to guarantee water access, as planning actions to achieve a safer city. Liu et al. (2014) analyze post-earthquake reconstruction, showing how an appropriate arrangement of public spaces and design of a network of open spaces and evacuation routes allows changing the urban form, from a dense spatial structure to one that is more attractive, safe, and resilient to disasters. The works of León and March (2014, 2016) emphasize the importance of urban design, especially the road network and public spaces, in

evacuation, the search for shelter and access to basic and emergency services in the event of a tsunami. Similarly, Ciborowski (1982) suggests that a design that considers the capacity and accessibility of the road network and the presence of evacuation routes can decrease urban vulnerability and have a significant impact on the behavior and attitude of individuals in the face of large seismic events.

Urban planning has the potential to facilitate the emergency response of coastal communities against tsunamis if geophysical

knowledge is properly integrated. Planning can have positive effects throughout all the stages of emergency management in case of an earthquake and tsunami event. During the catastrophe it can provide safe routes for evacuation and sheltering of evacuees and allowing emergency services to reach in-need populations, thus allowing the city to rapidly begin recovery processes (Allan et al., 2013). Good management of the urban form can reduce vulnerability and accentuate resilient characteristics in a city by creating better conditions for coping with alterations caused by events such as earthquakes and floods. To achieve

resilience in cities, it is necessary to generate redundancy degrees, which can be accomplished through increased spatial and functional diversity, and spatial integration of the ecosystem into urban planning (Allan et al., 2013). Proper urban design not only enhances urban preparedness and safety, but also promotes city development and tourism growth through a better use and improvement of public spaces (Liu et al., 2014).





Most of the aforementioned studies focus on proper urban management on the macro scale of urban configuration; i.e., the system of connected spatial elements that should be available to foster a prompt evacuation of the community at risk (Hillier et al., 1993); mainly through modifications in road connections, creation of open spaces, and guaranteeing the access to shelters and availability of basic services. León and March (2016) report on a gap in the literature regarding risk reduction. They suggest

that it is also necessary to carry out analysis from a micro-scale perspective, i.e., at the pedestrian experience level, of the public spaces available for evacuation and access to safe areas in case of events such as tsunamis. In addition, Reyes and Miura (2016) mention the importance of evaluating the susceptibility and reliability of evacuation routes with a detailed focus. In the light of the exposed literature, micro-scale urban vulnerabilities assessment and the identification of their potential negative impacts in evacuation processes is largely justified.

## 10  3   Tsunami hazard in Iquique

The Chile-Peru subduction zone, between the Nazca and South American plates, has an extremely high seismic activity, producing large earthquakes (Mw > 8) about every 10 years (Contreras-Reyes and Carrizo, 2011); some of greater magnitude have triggered tsunamis, causing loss of human life and substantial infrastructure damage (Lomnitz, 2004; Cisternas et al., 2012). Three large earthquakes have occurred between years 2010 and 2016 in the Chilean territory: the 27th February, 2010

Mw 8.8 Maule (Fritz et al., 2011), the 1st April, 2014 Mw 8.2 Iquique (Catalán et al., 2015; Tomita et al., 2016) and 16th September, 2015 Mw 8.3 Illapel (Aránguiz et al., 2016; Contreras-López et al., 2016). These earthquakes triggered tsunami waves that resulted in great damage to ports, coastal cities, and fishermen's coves.

Despite the magnitude of these earthquakes and their resulting tsunamis, the death tolls were low in comparison to lower-magnitude events such as the Mw 7.7 Mentawai Islands earthquake and tsunami with a death toll around 500 people (Satake

et al., 2013). The aforementioned authors attribute this fact to the fast self-evacuation culture promoted among the residents, demonstrating the importance of education, awareness programs and efficient warning (especially when contrasted with the delayed evacuation during 2010 Maule tsunami), showing that lessons from past events have been learned (Okal, 2015). Notwithstanding, to develop a safe self-evacuation it is required a spacial environment able to give an appropriate support and also withstand the previous earthquake.

The Norte Grande region in Chile (an area around 1,000 km long, comprising from the Arica Region to the south of the Antofagasta Region, Figure 1), has been the source of constant public and scientific concern due to the existence of a seismic gap that has produced an accumulation of elastic deformation (6 - 7 cm yr$^{-1}$) of high seismic hazard (Comte and Pardo, 1991; Métois et al., 2013). The last reported destructive tsunamis in this area occurred in 1868 (Mw~ 8.8), 1877 (Mw~ 8.8), and 2014 (Mw 8.2) (Hayes et al., 2014; Lay et al., 2014). Lower-magnitude earthquakes have also occurred in this area, such as

the 1967 Mw 7.4 and the 2007 Mw 7.7 events near the city of Tocopilla; however, a large part of the shallow subduction zone did not experience ruptures for a long period (Hayes et al., 2014). Indeed, the release of energy in 2014 proved to be less than expected, amounting to only around 20% of the total accumulated energy since the 1877 earthquake (Lay et al., 2014; Aránguiz





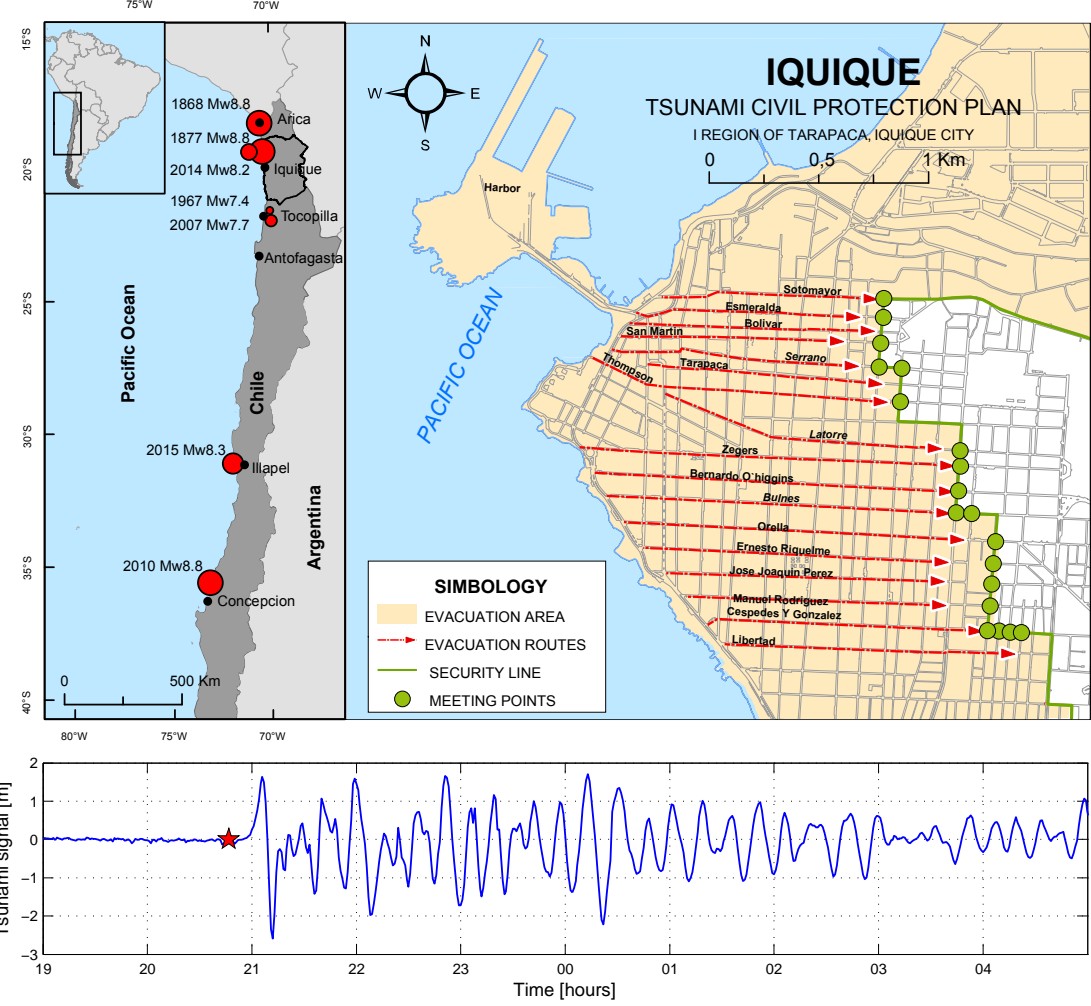

**Figure 1.** Upper left panel: Geographical location of the study area and a selection of major earthquakes (red circles) that have affected Central-North Chile since 1868 (epicenters based on the records of the National Seismological Center of the University of Chile and the United States Geological Survey). Upper right panel: Downtown Iquique and its tsunami evacuation routes (modified from (ONEMI and IMI, 2013)). Lower panel: The Iquique tide gauge tsunami signal for the 2014 event; the red star indicates the moment of earthquake occurrence.

et al., 2015). This suggests the possibility that a larger-magnitude event could occur in the future, the epicenter of which could be located either south or north of Iquique (Hayes et al., 2014).

The city of Iquique is located on the center of this seismic gap area (20.53° S, 70.18° W). Iquique's territory is a narrow coastal strip no wider than 3 km with a constant upward slope toward the east, where it is surrounded by a mountain system. Iquique is an important port and center of activity for the country's mining industry, with a population of 184,953 according





to the last census (INE, 2012). It plays an essential role in the transport of goods to nearby countries, supported by a growing duty free zone (which has also led to a boom in the city's car market and the subsequent high traffic levels).

Tsunami hazard has not been adequately recognized in the urban development of Chilean cities. Only after the 2010 tsunami the government amended the ordinance of urbanism and construction by defining restricted areas to urban development in 5 tsunami flood zones (MINVU, 2011), but there is still lack of national polices to mitigate the impact of tsunamis (Lunecke, 2015). Fortunately, Iquique has adequate urban morphological characteristics (on a macro level) for carrying out a fast evacuation, thanks to the orthogonal arrangement of its streets (León and March, 2016). This is a common factor in Latin American cities, which in coastal areas might lead to straight and redundant evacuation route layouts (Wyrobisz, 1980; Marshall, 2005), see Figure 1.

10 The 2014 earthquake triggered a moderate tsunami that resulted in minor flooding of the Iquique coast and nearby fishing villages, mostly damaging boats and small docks; no destruction of dwellings was reported and most damage was a result of the earthquake (Catalán et al., 2015; Tomita et al., 2016). In interviews done by Tomita et al. (2016), one of the stressed points was the importance of the drills conducted prior to the tsunami, which proved to be effective for identifying safe places and the closest evacuation routes at the time of the event. A large part of the population started a prompt evacuation after the end 15 of seismic shake and demonstrated a good understanding of tsunami warning issued by authorities (Tomita et al., 2016), which are proof of the ongoing evacuation education policies in the country. However, a range of problems were identified during the emergency, including the use of cars (which led to accidents and street blockages), and the lack of street lighting due to the massive power failure caused by the earthquake (León and March, 2016).

Records from the Iquique tide gauge (Figure 1) indicate that the first tsunami wave reached the coast at 20:56 (local time), 20 9 minutes from the nucleation of the earthquake; while the first peak, with a height of 1.6 m, was recorded at 21:06, thus only 19 minutes after the initiation of the seismic shaking (UNESCO/IOC, 2014). In addition, arrival times for the 2015 Illapel tsunami, were found to be less than 12 min in the field survey conducted by Aránguiz et al. (2016). This demonstrates that, depending on the location of the tsunami generation area, little time may be available for evacuating to safe areas; therefore, a rapid response and evacuation are essential as protective measures against near field tsunamis in Chilean coastal cities.

## 25  4   Methods

Public spaces (especially escape routes) have a critical role in case of a near-field tsunamis, by fostering the evacuation of pedestrians to safe areas. Ideally, these spaces should remain clear and free of obstacles in order to guarantee that the design capacity of the route is not altered (Scheer et al., 2012). In the fieldwork carried out by León and March (2016), a series of vulnerable points on a micro-scale level were detected on evacuation routes in Iquique, which were classified into three 30 categories: i) precarious physical conditions and inadequate maintenance, ii) problems related to the design of the public space, and iii) inappropriate use of sidewalks.

As a complement to the work of León and March (2016), we conduct here a detailed micro-scale analysis of the Iquique's urban context, as an attempt to characterize potential difficulties to carry out effective evacuation processes. The central part





of Iquique (see Figure 1), is an urban area characterized by a high population density and traffic flow, along with an intensive commercial, touristic, industrial, and educational activity; these factors contribute to a high exposure to tsunami hazard and motivate the development of the proposed methodology.

This research was developed following sequential trajectories (Cullen, 1961; Clay, 1994) of analysis along the evacuation

routes proposed by the municipality of Iquique in conjunction with the National Emergency Office in its plan for civil protection against tsunamis (ONEMI and IMI, 2013). These routes were defined as the shortest paths, oriented from west to east from the coastline that lead to high ground areas (30 meters above sea level). The methods used in this study comprise three steps. First, extensive fieldwork is conducted to perform a diagnosis of evacuation routes, following the path of evacuees during their escape. Next, surveyed micro-vulnerabilities are geo-referenced and classified according to their complexity and consequences

in evacuation, and finally, a friction rate that accounts for velocity reductions of pedestrians is proposed based on the literature review.

## 4.1   Fieldwork

During October 2015, a detailed diagnosis of the current state of evacuation routes was carried out through a fieldwork in the central part of Iquique. Specifically, the analyzed area is bounded from north to south by Sotomayor and Libertad streets

(see Figure 1), which includes the historic district and the area of influence of the port, alongside residential, educational and commercial activities. More than 45 kilometers of evacuation routes were assessed with the aid of video footages and Global Positioning System devices (GPS), which were used to geo-reference existing urban micro-vulnerabilities that pedestrians could experience during an emergency evacuation.

During the fieldwork, the following types of micro-vulnerabilities were observed as the most common elements capable of

hindering evacuation: i) presence of parked cars on sidewalks, ii) narrowing of sidewalks to make space for parking, iii) use of sidewalks to extend the service area of restaurants (only during the day and evening), iv) use of public spaces for informal commerce, and v) road works. The last is a temporary type of vulnerability; therefore, it represents a specific, non-regular condition in the city streets (see Figure 2).

Among the identified issues, that of greatest concern is the presence of parked cars on sidewalks. Due to the large dimensions

of these obstructions, the useful walkable area is reduced and the capacity of the evacuation routes is considerably affected; in some cases, the available width of the sidewalk is reduced to less than a meter, which in the event of an evacuation might lead to bottlenecks that could increase evacuation times. In addition, there are built parking areas that decrease the sidewalk width, called narrowings in this work. The presence of these elements is related to the public need for parking spaces due to the high motorization rate in the city, which is among the top 3% of Chilean communes with the highest number of vehicles (the

value of which is comparable with communes in the capital of the country (INE, 2015)). It was observed that a large portion of households, mostly old buildings, do not have private parking spaces, forcing the residents to park their vehicles in public spaces.



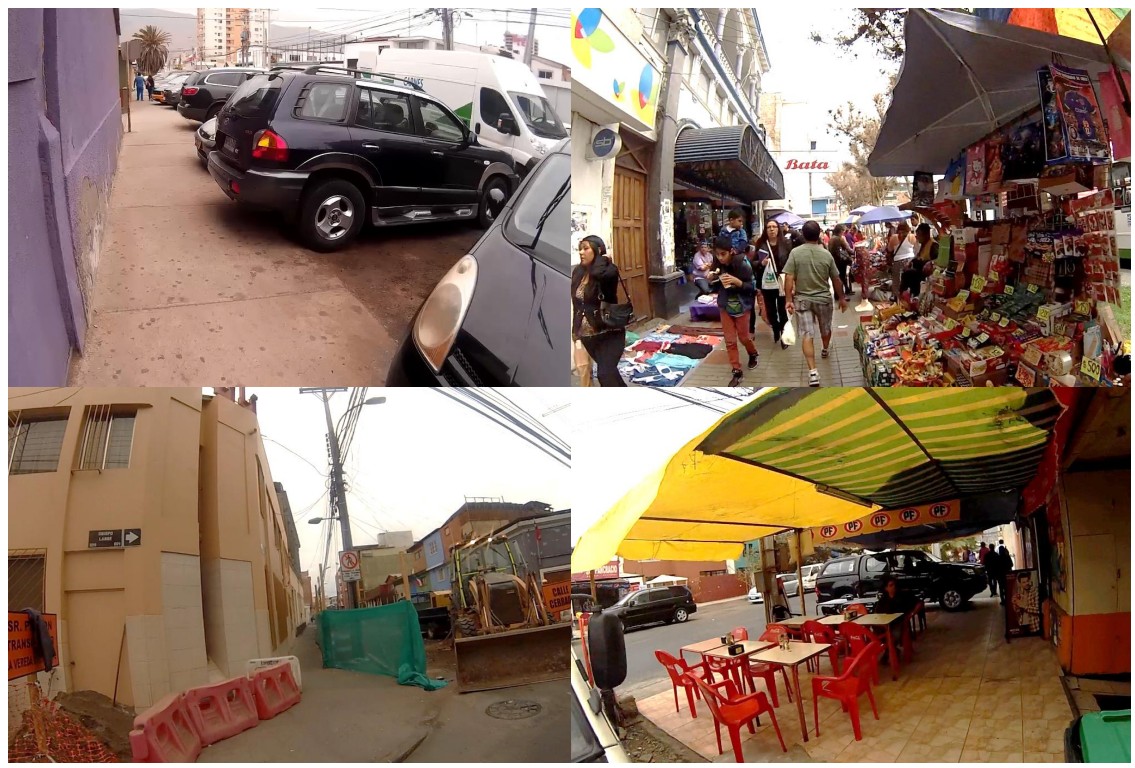

**Figure 2.** Still frames obtained from field recordings in the city Iquique. Improperly parked vehicles (top left), informal commerce (top right), road repairs (bottom left), restaurant tables on the sidewalk (bottom right).

## 4.2 Data analysis

During the post-processing stage, elements that represent an impediment or delay to pedestrians movement along evacuation routes, thus contributing to the vulnerability of public spaces, were characterized through a thorough analysis of the fieldwork-collected data. This process consisted in the identification of the elements within the studied urban spaces that could decrease

5  the speed of evacuees, considering the results from the literature on pedestrian dynamics Soule and Goldman (1972); Fujiyama and Tyler (2004). Subsequently these elements were classified according to three principal criteria: i) blocking or decrease in spaces available for movement, ii) abrupt surface level changes, and iii) considerable changes in surface roughness.

After the micro-vulnerabilities were identified, they were drawn on the map of the city using geographic information tools (ArcGIS) in a planar projection system, attempting to faithfully reproduce their dimensions in order to generate a map that

10  includes the micro-vulnerabilities existing along each of the evacuation routes in the study area. The mapping of micro-vulnerabilities using geographic information systems delivers information regarding the characteristics of the element, its location and the surface area it covers, facilitating the organization, manipulation and analysis of the large quantity of data obtained (Figure 3).



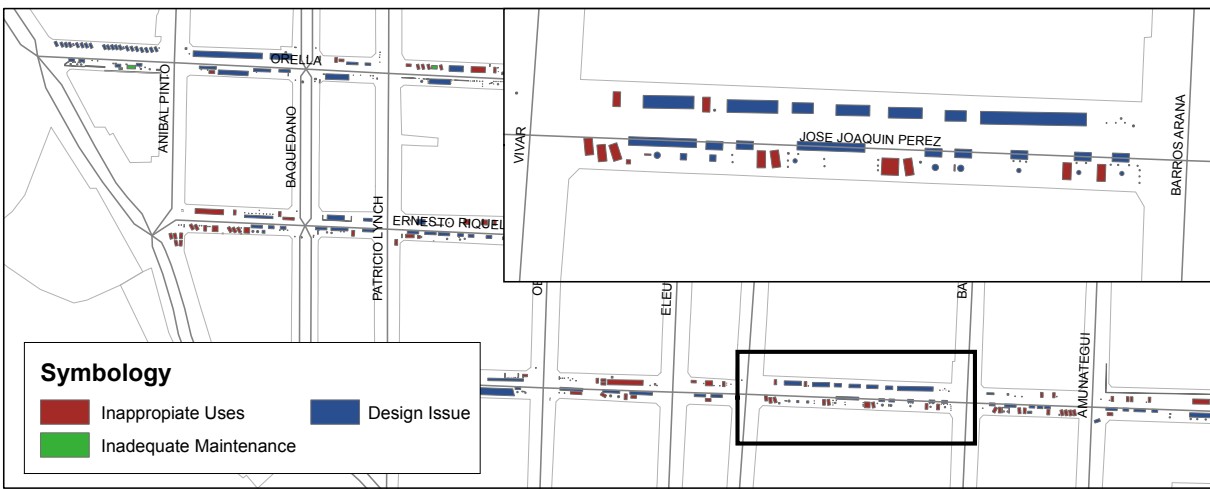

**Figure 3.** Mapping of identified micro-vulnerabilities, colored according to their taxonomy.

**Table 1.** Classification based micro-vulnerability origin.

| Taxonomy | Description |
| --- | --- |
| Inappropriate use | Unsuitable use and appropriation of the sidewalk for multiple uses other than pedestrian traffic, e.g., parking, restaurants, gardens. |
| Poor maintenance | Lack of care and repair of public spaces on the part of the relevant authority or individuals, e.g., broken sidewalks and manhole covers. |
| Design problems | Difficulties present on routes associated with their planning and construction, e.g., sidewalk narrowings and stairs. |

In order to generate a taxonomy based on the origin of the observed problems, following the guidelines proposed by León and March (2016), they were grouped into three categories: i) inappropriate use, ii) inadequate maintenance, and iii) problems related to the design of evacuation routes, the descriptions of which are found in Table 1. This classification follows a representation of the micro-vulnerabilities in terms of their origin, and also provides an indication regarding policies or regulations

5  that could be implemented at the municipal level to decrease their effects during future evacuation processes. While the inappropriate use and inadequate maintenance of routes can be rectified through easy-to-implement strategies, the problems related to design would need more invasive measures.





### 4.3 Evacuation route obstruction level

In the previous section, the types of difficulties that a pedestrian could encounter while evacuating along the routes of Iquique were described qualitatively. Our research also provides tools to evaluate quantitatively the detected problems and compare between evacuation routes. The quantification of the micro-vulnerabilities and the obstruction levels of the evacuation routes are defined through a proposed friction rate, defined as:

$$i[\%] = \frac{\sum_j S_{m_j} \cdot \alpha_j}{S_r} \times 100 \qquad (1)$$

$$\alpha_j = 1 - SCV_j \qquad (2)$$

Where:

$S_m$: Surface area of the micro-vulnerability associated with an evacuation route.

$S_r$: Surface area of the analyzed evacuation route.

$\alpha$: Speed reduction factor associated with each micro-vulnerability.

This indicator represents the proportion of the area of an evacuation route that is occupied by the micro-vulnerabilities existing on it. To distinguish how pedestrians are affected when confronted with a given micro-vulnerability, the factor $\alpha$ is defined to quantify their speed reduction; therefore, the effect of each element is weighted differently in the friction rate. The factor $\alpha$ is the complement of the magnitude defined as Speed Conservation Value ($SCV$), Equation 2 (Wood and Schmidtlein, 2012; Schmidtlein and Wood, 2015), which represents the percentage of the maximum speed that can be maintained on a given surface. To this end, each micro-vulnerability was classified according to one of these categories: i) blockages, ii) level changes, and iii) surface roughness. Where the maximum speed is reached on compacted and flat ground such as street pavement and sidewalks (Soule and Goldman, 1972), in which cases speed is completely conserved $SCV = 1$.

For each micro-vulnerability, a $SCV$ value was assigned (see table 2). For elements where passage through is not possible and thus represent a blockage of pedestrian movement, the value of $SCV$ is null. Among the observed micro-vulnerabilities, there are some which do allow passage, but involve a change in the normal speed of movement such as level changes that require pedestrians to make an additional effort to continue onward, and surface material changes that translate into more difficult movement. The chosen values were selected from literature regarding pedestrian speed measurement in various situations; in the case of level changes, the speed conservation value was defined as around 50% (Fujiyama and Tyler, 2004) and for surface material changes, around 90% (Wood and Schmidtlein, 2012; Schmidtlein and Wood, 2015).

In summary, Equation 1 represents the sum of all the areas of micro-vulnerabilities on a particular evacuation route, individually weighted by a speed reduction factor based on experimental literature review, Equation 2. The friction rate is the quotient of this sum and the total surface area of the evacuation route, which is then multiplied by 100 in order to work in percentage





**Table 2.** Speed conservation of micro-vulnerabilities present on evacuation routes.

| Speed Conservation Value (**SCV**) | | | |
|---|---|---|---|
| Blockages (**0**) | | Level changes (**0.5501**) | Surface roughness (**0.9091**) |
| Barriers | Narrowings | Manhole covers | Cracked sidewalks |
| Basement entrances | Objects | Stairs | Green areas |
| Bus stops | Parking spaces | Uneven surfaces | |
| Cars | Public telephones | | |
| Debris | Restaurants | | |
| Electrical boxes | Road separators | | |
| Electricity pole | Road works | | |
| Fences | Sculptures | | |
| Fire hydrants | Seats | | |
| Garbage cans | Signs | | |
| Gas stations | Trees | | |
| Informal commerce | Walls | | |
| Kiosks | | | |

Based on Fujiyama and Tyler (2004); Wood and Schmidtlein (2012); Schmidtlein and Wood (2015)

terms. The analyzed area comprises evacuation routes with similar road dimensions along its length; therefore, the total evacuation route surface area is used, making it possible to analyze the contribution to vulnerability of cars that are completely or partially parked on the road. In cases in which the dimensions of the road's cross section varies among the analyzed evacuation routes, separate analyses of the sidewalk and street may be done.

## 5  Results and discussion

The purpose of calculating the friction rate at evacuation route level as defined in the previous section is to have a measurement of the effect of micro-vulnerabilities on the available space for the movement of evacuees along each evacuation path, allowing a comparison of them and to determine their relative degree of vulnerability. Providing a useful tool to prioritize focus zones and develop more accurate evacuation models in these areas.

This index includes the sidewalk and street surfaces to represent the obstruction level of the evacuation routes, but it removes the cars circulating on the street from the identified micro-vulnerabilities, due to the variability of this condition. The high motorization rate in the city and the history of vehicles used in previous evacuations (Matus and Muñoz, 2014; Riveros, 2014) pose a negative precedent and undoubtedly increase the vulnerability of people during an evacuation, as they bound movement of pedestrians mostly to the sidewalk (see Figure 4).




**Figure 4.** Preventive evacuation of 16 March 2014 (CNN Chile, 2014).

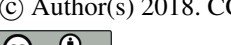



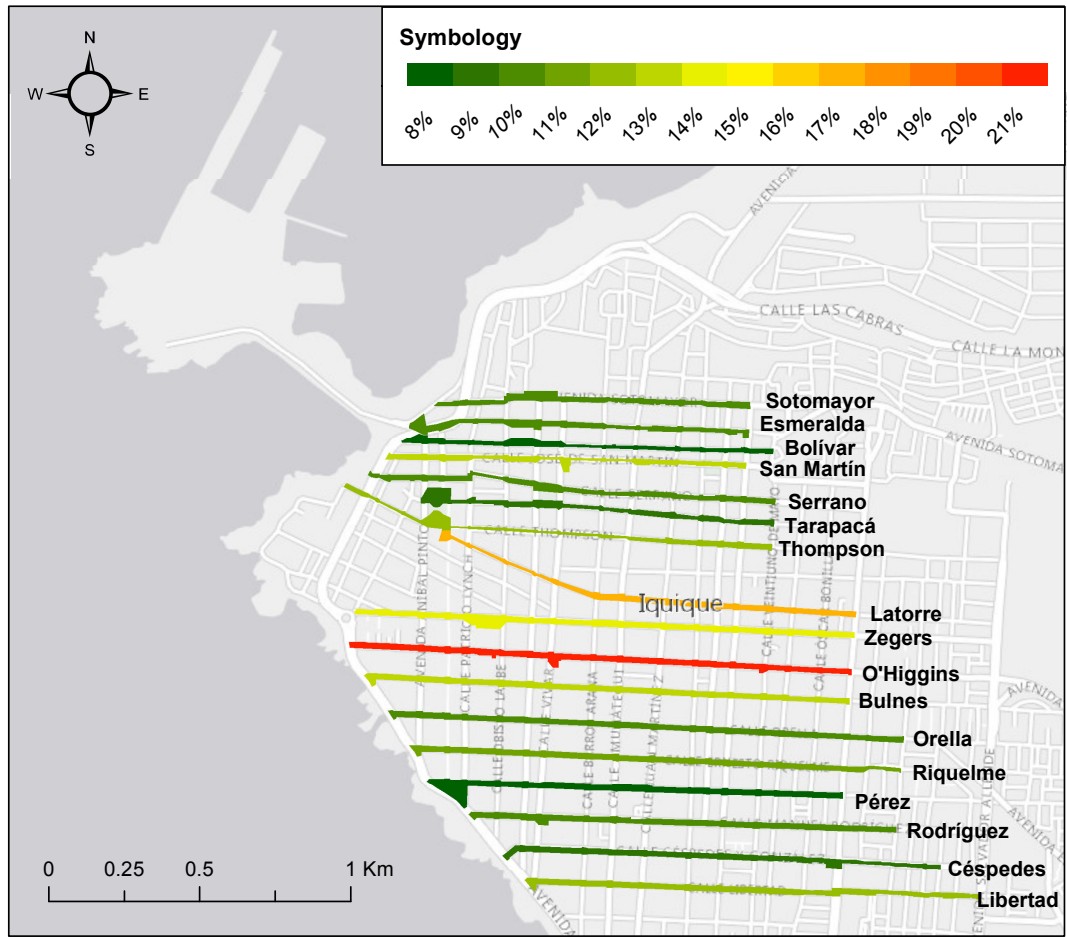

**Figure 5.** Friction rates obtained for the evacuation routes in the study area.

In Figure 5, the results of the friction rate calculations for the streets analyzed in this study are presented graphically. It is showed that the highest friction rates are concentrated in the central part of the study area, due to its status as the most active zone of Iquique's downtown, which is result of the high presence of educational institutions, office buildings and commercial facilities. The maximum friction rate value, for O'Higgins Street (an evacuation route that in some stretches at the moment of the study was affected by road work) verges on 20%. The second route with a high friction rate, Latorre Street, is object of concern due to the quantity of people that would arrive during an evacuation, especially since it provides the shortest means of access to the safe area for part of the population that resides on the peninsula south of the port (around 530 people). Continuing with this analysis, the rest of the population of the peninsula (around 780 people (INE, 2012)), must evacuate by Zegers Street, the third evacuation route that has a large number of micro-vulnerabilities along its entire length.



The port area deserves a special attention due to its large distance to safe areas. In this respect, representatives of the Iquique Port estimated that in a worst case scenario there could be more than 1,500 workers. Fortunately, the evacuation routes accessible by the Iquique port's personnel, Esmeralda and Bolívar streets, have low friction rates. However, the greatest problem for workers is the time it takes to leave the port and reach the mainland safe area. Considering the aforementioned observations,

the major concern in Iquique downtown is Zegers Street by its high friction rate and nearness to coast unlike Latorre Street; on the other hand, O'Higgins Street must be removed from this analysis due to the temporal elements contributing to its friction rate during the fieldwork. The rest of the evacuation routes show lower friction rates, but some speed reductions and flow capacity decreasing of evacuation routes are to be expected. By gathering all the micro-vulnerabilities and adding its contribution to the friction rates in the study zone, it is possible to deduce that the main issue is the inappropriate use of

urban space to parking on either the sidewalk or street, with around 74% of the whole friction contribution. One of the most illustrative outcomes of the developed methodology is to provide means to assess the drawbacks of using sidewalk for parking, generally perpendicular to the road, with the effective width of the sidewalk sometimes decreasing from 4 to practically 1 m. Through the presented methodology of micro-vulnerabilities identification is possible to pinpoint urban problems related to evacuees displacement, quantify its impacts and prioritize solutions with different complexity levels.

In Chile the definition of threatened and tsunami safe areas, along with the elaboration of evacuation plans is duty of National Emergency Office (ONEMI), while the Ministry of Housing and Urbanism (MINVU) is the urban planning and development manager, who also defines the use of urban spaces. Nevertheless, the execution and supervision of higher authorities plans is in the hands of municipalities. This institutional fragmentation leads to different urban risk reduction approaches, despite the above, efforts haven been made to standardize the urban planning procedures related to tsunami evacuation infrastructure

(Gutierrez et al., 2016).

An interview with personnel from the regional branch of the National Emergency Office and the municipality of Iquique was conducted as part of the field survey, where the plans related to evacuation improvement were discussed. The aforementioned authorities put forth ideas for improving evacuation plans and infrastructure, among which was the creation of main routes intended exclusively for pedestrian use during an evacuation. Internalization of evacuation procedures among the population

was highlighted and concerns about the use of cars during the last evacuation process in 2014 was also mentioned during the interview. As result of this study a series of mitigation measures that could be carried out at the municipal level in Iquique, are proposed in order to improve the urban design for evacuation. The most direct of which is the regulation of parking in public spaces, by using city council attributions like fines in areas near to the coast. Other measures like the creation of additional parking spaces on streets perpendicular to evacuation routes, installation of elements that impede the passage of cars onto

the sidewalk and removal of abandoned vehicles, are strongly recommended. Likewise, temporary activities such as informal commerce and dining must take place only on streets not meant to be used as evacuation routes. The implementation of such modifications must take into account possible effects on existing traffic and the local economy as well as socio-political reactions.

The presented results demonstrate the existence of urban micro-vulnerabilities in the study area and characterize their effect

in tsunami evacuation processes. However, the fieldwork was undertaken during a specific time window and no time evolution





analysis of the locations of the micro-vulnerabilities on the streets of Iquique was carried out. Updating the friction indexes would require a semi-continuous survey of the evacuation streets, which could be performed using for instance Closed-Circuit Television (CCTV) systems available at the municipal level. Doing a temporal analysis could be possible through automated micro-vulnerability mapping processes using tools for object detection in images as well as satellite images of extremely high

resolution and capture frequency. Notwithstanding the foregoing, the present study examines a situation as close as possible to the everyday nature of the city, i.e., during a non-holiday time of the year and day of great work, educational and commercial activity. The presented methodology is an important step to characterize urban vulnerabilities at micro-scale, quantify its effects and providing tools for decision makers related to urban design and improvement of evacuation processes.

## 6 Conclusions

Tsunamis are one of the natural hazards of most concern for coastal cities and port areas. Specially in Chile, whose maritime border is close to one of the most active tectonic plates. To mitigate their potential impacts, joint efforts and actions are needed, including political, technical, financial, and cultural. Tsunami education and evacuation drills programs had good results and demonstrated to be effective during the 2014 and 2015 Chilean tsunamis (Aránguiz et al., 2015; Tomita et al., 2016), however these events had lower magnitude than the 2010 tsunami disaster. The last two events have allowed to assess the response of

coastal cities against lower magnitude events, and to develop new strategies for the continuous improvement of the evacuation response against major tsunamis. Improving the environment and the ability of the community to adequately react when confronted with a great disturbance is fundamental, since, limited preparedness gives rise to vulnerable environments (Guha-Sapir et al., 2011; Yi and Yang, 2014).

This article highlights the importance of evacuation routes, public spaces intended to multiple uses that also support tsunami

evacuation, and the assessment of the built environment condition. It is evaluated the presence of hindering physical features resulting from planning, use and maintenance problems, and their influence in the displacement of people to safe areas, therefore delaying evacuation processes. The decrease of the effectively available pedestrian area in evacuation routes caused by the presence of sidewalk obstructions was identified as the greatest problem in the city of Iquique, which could worsen over time due to the higher activity and population growth trend in coastal areas. When this growth occurs in conjunction with urban

improvements in pursuit of resilience on both the macro and micro scales, it promotes better prepared environments and could improve the people response in case of emergencies. The construction of high-rise buildings has increased lately; the possibility of using them as vertical evacuation shelters must be carefully analyzed, taking into account their seismic design, capacity and resistance to hydrodynamic forces. The availability of buildings for vertical evacuation purposes would considerably decrease the flow of pedestrians during an evacuation, promoting safer evacuations. In particular, their use is strongly recommended

for evacuating workers from the port of Iquique (Solís and Gazmuri, 2017; León and March, 2016), which represent the city's district most exposed to tsunamis, due to its distance from the safe high areas.

As mentioned in section 5, the main issue in Iquique is related to cars use. The contribution of other micro-vulnerabilities to the friction rates and their effect in the decreasing of flow capacity of evacuation routes is lower, with the presence of





informal businesses, restaurant expansions and trees on the evacuation routes, which have values between 2-3% of the whole friction factor. For the purpose of decreasing the existence of micro-vulnerabilities, measures can be taken at the local level by municipal authorities. The inappropriate use of evacuation routes can be rectified through restrictions and fines designated by the municipality. Likewise, methods can be developed to monitor the possible public space difficulties generated by inadequate

route maintenance. Finally, problems related to design are the most difficult to reverse and can be corrected through the construction of new works or retrofitting of existing areas, with evacuation efficiency taken into account as an important driver.

While the road network and the infrastructure of a city play an important role during an emergency, it is also possible for a series of difficulties to emerge after an earthquake, for instance the failure of power supply, and the built environment precarious condition as a result of the strong earthquake, in addition to existing micro-vulnerabilities. Generally, the identification of urban

micro-vulnerabilities is not been carried out on large scale, and investment on mitigation measures is not a main concern in developing countries. In near field tsunami prone areas, where the available time to evacuate could be very short, like Chilean coastal cities, urban micro-vulnerabilities could make the difference in the evacuation process performance. If the evacuation routes remain clear and free of obstacles, the evacuees movement could be faster and the overall tsunami evacuation time could decrease and eventually save lives. The assessment of evacuation routes condition should be done regularly, specially in South

Pacific countries with high seismic and tsunami risk, where the environment showed by maps and plans is commonly different from the reality that pedestrians experience.

Efforts to create a more resilient city must not be made only after a disaster, instead should be incorporated in the decision-making process of city planning. Receptiveness to information delivered by the authorities in order to avoid incorrect conduct and to foster appropriate use of public spaces are undoubtedly important everyday behaviors that decrease the vulnerability of

the population and contribute to higher resilience of coastal zones.

*Acknowledgements.* This research was supported by the Research Center for Integrated Disaster Risk Management [CONICYT/FONDAP/15110017]. The authors are grateful for the information provided by the Iquique Municipality and the Tarapacá office of Chilean Emergency Management Agency, ONEMI.





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
