# Peer review of "Identification and classification of urban micro-vulnerabilities in tsunami evacuation routes for the city of Iquique, Chile"

_Natural Hazards and Earth System Sciences, 2017_

## Referee Comment (RC1) · Dr Gonzalez-Riancho (Referee) · 3 Mar 2018

General comments:

This paper addresses the identification and classification of urban micro-vulnerabilities affecting the capacity of evacuation routes and hindering evacuees 'safety. The method is applied to the city of Iquique, which has a population of around 185,000 inhabitants and is located in a tsunami-prone area affected by tsunami events in the past.

The paper includes an introduction, a theoretical background, a description of the study area, a description of the methodology, a discussion of the main results, as well as

conclusions. The article is well written and based on an in-depth understanding of the evacuation's main concepts. The methodology applied is based on a previous work by Leon and March (2016) in Iquique, which is complemented here with a detailed micro-scale analysis of the same city. The main contribution of the paper is a method to prioritize evacuation routes based on the calculation of friction rates and the effect in the decreasing of the evacuation flow capacity of the routes. The method and the conclusion obtained from its application to Iquique prove that the research is highly useful for evacuation planning and for the rethinking of the cities 'urban design in post-disaster reconstruction phases. It can be easily replicated to other areas and by future researchers or coastal practitioners.

Specific comments:

The method applied includes three steps: (i) diagnosis of evacuation routes, (ii) geo-referencing and classification of micro-vulnerabilities, and (iii) calculation of the friction rate. The titles of the various sections in the method chapter would give a clearer idea of the work if they mention the method steps instead of generic terms valid for any scientific study as "fieldwork" and "data analysis".

The work is based on a previous publication by Leon and March (2016) as mentioned in Page 6, and a great part of the method applied here follows that one. At least two of the three steps (diagnosis of the evacuation routes and classification of micro-vulnerabilities) seem to be based on criteria and categories defined in Leon and March (2016). This reviewer has not been able to have access to the work by Leon and March (2016), and consequently could not prove if the work presented by the authors is original or if it is a case study or a method already published. It is crucial that the authors clarify which parts from their work are original and which ones are not. The main differences between the two works carried out in Iquique should be clarified to better understand if there are scientific innovations in this work or if it is a case study applying the method from Leon and March (2016).

In the Data analysis Section, two different classifications of the elements found in the evacuation routes are described. The first classification, based on (i) blockages, (ii) level changes, and (iii) surface roughness, seems a bit disconnected to the method described in pages 8-9. Only after reading the next section on friction rates (pages 10-11) the role of this classification is understood. It would be advisable to mention in page 8, lines 5-7, that this classification is used later for the calculation of the friction rates.

Technical corrections:

• Page 2, line 9: replace ". . .at the coast Mori et al. (2013); Fraser et al. (2013)" with "at the coast (Mori et al., 2013; Fraser et al., 2013)" • Page 4, line 19: please add the year of the Mw 7.7 earthquake and tsunami in Mentawai Islands. 2010? • Page 4, line 23: replace "spacial" with "special" • Page 9, Table 1: Leon and March (2016) should be cited in the table caption.

---

## Short Comment (SC1) · 12 Apr 2018

We thank Dr. González-Riancho Calzada for providing feedback for improving our manuscript. Regarding her specific comments:

1) She points out that "The titles of the various sections in the method chapter would give a clearer idea of the work if they mention the method steps instead of generic terms valid for any scientific study as "fieldwork" and "data analysis"". We agree with this comment. The improved version of the manuscript will include more accurate titles for each section, as it is currently the case of section 4.3: "Evacuation route obstruction level".
2) Dr. González-Riancho Calzada also underlines that, regarding previous publication by León and March (2016), "It is crucial that the authors clarify which parts from their work are original and which ones are not. The main differences between the two works carried out in Iquique should be clarified to better understand if there are scientific innovations in this work or if it is a case study applying the method from Leon and March (2016)". We agree that this work is referenced several times throughout the new manuscript (which may confuse some readers about the the originality of this). However, we also point out that León and March's document introduce a more qualitative approach to the diagnosis of evacuation routes and classification of micro vulnerabilities. Indeed, León and March (2016) underline the need for a more quantitative analysis for these issues, which is the starting point of this new manuscript. We will clarify this issue on the improved version of our manuscript. We also attach here León and March (2016) original paper.

3) The reviewer also points out that "In the Data analysis Section, two different classifications of the elements found in the evacuation routes are described. The first classification, based on (i) blockages, (ii) level changes, and (iii) surface roughness, seems a bit disconnected to the method described in pages 8-9. Only after reading the next section on friction rates (pages 10-11) the role of this classification is understood. It would be advisable to mention in page 8, lines 5-7, that this classification is used later for the calculation of the friction rates". We agree with this comment. Content of page 9 is focused on introducing a classification scheme according to each micro-vulnerability's cause, not necessarily related to the characteristics of the micro-vulnerability itself (which in turn are the inputs for friction rates). We will clarify this issue in the improved version of our manuscript.

Again, we thank Dr. González-Riancho Calzada for her comments, and we believe that the new version of our manuscript will be a significantly improved one and more readable for broader audiences.

Please also note the supplement to this comment:
https://www.nat-hazards-earth-syst-sci-discuss.net/nhess-2017-458/nhess-2017-458-SC1-supplement.pdf

———————————————————————

[Figure]

**Supplement:**

[Figure]

Planning and Design

Environment and Planning B:
Planning and Design
2016, Vol. 43(5) 826–847
© The Author(s) 2015
Reprints and permissions:
sagepub.co.uk/journalsPermissions.nav
DOI: 10.1177/0265813515597229
epb.sagepub.com

**$SAGE**

**An urban form response to disaster vulnerability: Improving tsunami evacuation in Iquique, Chile**

**Jorge León**
Faculty of Architecture, Building and Planning, University of Melbourne, Melbourne, Australia;
Departamento de Arquitectura, Universidad Técnica Federico Santa María, Valparaíso, Chile

**Alan March**
Faculty of Architecture, Building and Planning, University of Melbourne, Melbourne, Australia

**Abstract**
As urbanization gathers pace and climate change increases the number and magnitude of many natural hazards, cities are increasingly becoming hot spots for disasters. Although the role of appropriate urban forms in reducing disaster vulnerability has been recognized for some time, the majority of its potential remains focused on long-term mitigation efforts. In contrast, examination of the relationships with short-term disaster management activities such as response and immediate recovery has not been thoroughly conducted. This paper contributes to this shortfall by analysing a critical type of rapid onset disaster, a near-field tsunami, and the role of urban form in supporting the populations' core response activities of evacuation and sheltering. The Chilean city of Iquique (affected by a severe earthquake and minor tsunami in 2014) is examined using a mixed methods approach that provides the basis for proposed macro-scale and micro-scale changes in its urban form; these modifications, in turn, are assessed with geographic information system (GIS) and agent-based computer models. The results show important existing evacuation vulnerability throughout major areas of the city (as the result of interrelated critical conditions), which nonetheless could be significantly reduced by the changes proposed. Further steps in this iterative process, in turn, could lead to the development of evacuation-based urban design standards capable of being transferred to different tsunami-prone contexts around the world.

**Keywords**
Urban form, 'what-if' scenarios, tsunami, evacuation, agent-based model

**Introduction**

To minimize their overall impacts, rapid onset urban disasters like near-field tsunamis require prompt responses from vulnerable populations, including appropriate decisions

**Corresponding author:**
Jorge León, University of Melbourne, 757 Swanston Street, Parkville, Melbourne, Victoria 3010, Australia.
Email: jleon@student.unimelb.edu.au

about critical activities such as evacuation and sheltering. Usually the people conduct these activities without official guidance and in a hazardous urban environment, due to 'cascading failures' brought about by the previous large-magnitude earthquake. It has been argued that the characteristics of urban form can increase a community's capacities to deal with unfolding crises like this and therefore achieve urban resilience to disasters (Allan et al., 2013); nevertheless, little literature exists regarding this link (Allan et al., 2013; Allan and Roberts, 2009) and changes that might bring about improvement are not commonly examined. In this respect, the majority of tsunami risk-reduction efforts examine the built environment (e.g. land zoning) and emergency readiness actions (e.g. evacuation preparation) as separate areas of study and practice: whilst the former focuses on long-term changes excluding evacuation analysis, the latter examines the urban realm as a static context. This paper analyses this gap in the tsunami-prone city of Iquique, Chile, by including a diagnosis of the existing situation, a series of urban modifications to improve these conditions and an appraisal of these changes by qualitative and quantitative tools.

   The paper is divided into five parts. First, a theoretical background about disasters (particularly tsunamis) and the role of urban form as a coping strategy is set out. Second, the case study of Iquique is introduced, including the proposed research method. Third, the existing situation and urban recommendations for improvement are examined with the aid of computer-based models and fieldwork. Finally, the results and their implications for tsunami risk reduction in the urban built environment are discussed.

**The role of urban form in coping with disasters**

**From emergency management to disaster risk reduction**

The global number of yearly reported natural disasters has quadrupled since the 1960s, reaching more than 400 events and 200,000 affected people per year in the 2010s (International Disaster Database, 2011). This growth can be partially explained by interrelated factors such as global warming and poor management of natural resources (Joerin and Shaw, 2010). Nevertheless, probably the most important factor is the increasing exposure of people to natural hazards, caused by human development patterns such as rapid urban growth and rising social inequalities, especially in the developing world. Moreover, as urbanisation gathers pace, cities have become 'hot spots' for disasters (Joerin and Shaw, 2010; Wamsler, 2014).

   Since the 1950s, governmental tactics for coping with disasters have evolved from a civil-defence-based response and relief approach to a risk-reduction strategy (Pearce, 2003; Tarrant, 2006; UNISDR, 2004). Within this framework, the potential role of built environment disciplines such as urban planning and urban design has been increasingly accepted as integral to the long-term mitigation of hazards and location-specific vulnerabilities. This can be achieved via the management and modification of the spatial arrangements, functions and ongoing growth or decline of cities and regions (Burby et al., 1999; Wamsler, 2014).

**Urban form and resilience to disasters**

Alongside long-term risk reduction, appropriate urban forms can also improve other disaster management actions, such as response and recovery. Allan et al. (2013) suggest that urban morphology might increase a community's capacity to quickly and effectively respond to major disturbances like earthquakes. This can be achieved, for instance, by providing a network of open spaces (streets, parks, squares, etc.) capable of accommodating response and recovery activities such as shelter, health care, distribution of goods and services, disposal of waste, commerce, etc. (Allan and Bryant, 2010; Allan et al., 2013). These

actions are essential for attaining urban resilience to disasters, i.e. the ability to cope with these catastrophes by surviving them and minimizing their impacts, resulting in recovery with minor social disruption (Cutter et al., 2008).

Despite the critical importance of urban form in achieving resilience, little literature exists regarding this connection (Allan et al., 2013; Allan and Roberts, 2009). As Allan et al. (2013: 244) argue, 'recovery planning and emergency management documents typically refer to the urban environment as a place that should be recovered rather than one that might support recovery'. Noticeable exceptions to this gap are authors like Cai and Wang (2009), Ciborowski (1982), He and Xu (2012), Ishikawa (2002) and Wu (2012), who have examined the role of urban form during a range of disaster response activities like evacuation, sheltering, crowd behaviour and accessibility of emergency relief services to vulnerable areas.

**A critical case of rapid onset disasters: Near-field tsunamis**

The importance of urban forms appropriate to the risks presented by hazards is especially critical in the case of rapid onset disasters such as near-field tsunamis affecting densely populated coastal locations. In cases like this, the vulnerable population has little time (typically only minutes or few hours) to make appropriate decisions about essential activities such as evacuation and sheltering, which in turn have a strong influence on the overall impact of the catastrophe. Moreover, in many occasions these actions have to be autonomously conducted by the people, due to the cascading failure of emergency systems following a strong earthquake (Alesch and Siembieda, 2012). This context of crisis can be alleviated by the urban form having characteristics that promote resilience in the form of support for rapid and effective response.

A literature review of theoretical and applied tsunami risk reduction efforts in coastal communities (Bernard, 1995; Kazusa, 2004; Murata et al., 2010; Preuss, 1988; Scheer et al., 2012; Shuto and Fujima, 2009) shows that there are three main types of strategies for coping with tsunami risk in coastal areas: (1) extensive civil-engineered defences (e.g. breakwaters, sea gates, seawalls, control forests); (2) land use and built environment measures (e.g. zoning, building codes and design recommendations) and (3) emergency readiness systems (e.g. education, warning systems and especially population evacuation preparedness). Given that large-scale engineered countermeasures have not been widely used outside Japan, tsunami evacuation remains 'the most important and effective method to save human lives' (Shuto, 2005: 8).

Regarding built environment-based strategies, Carmona et al. (2010) and Conzen (1960) point out that there are four essential urban form elements: (i) streets (arranged in a street system), (ii) plots (aggregated in street blocks), (iii) land uses and (iv) building structures. These elements are part of the two key features of any urban tsunami evacuation system: routes and safe assembly areas (either 'horizontal', i.e. open public spaces, or 'vertical' e.g. existing buildings or evacuation towers). The disciplines that foster appropriate urban forms, like urban planning or urban design, can therefore have a strong influence on the overall outcome of an emergency evacuation. Within tsunami risk reduction frameworks, urban form and evacuation should be strongly linked in a two-way relationship subject to ongoing review and improvement. Nevertheless, current approaches, upon examination, typically exhibit a lack of connection between these two aspects.

Urban form-related tsunami risk reduction approaches (such as zoning, relocation of population and activities, creation of buffer zones, and design or building codes) are long-term mitigation strategies. It is common that a deep analysis of actual populations' responses in the case of tsunami emergency is not undertaken when these approaches are employed.

In turn, evacuation analyses commonly examine the different aspects of recorded or potential crises (e.g. population behaviour, involved departure/displacement times, risks, etc.) but with an exclusive focus on the description and/or diagnosis of the situation, without a deeper examination of how the urban spaces themselves could be designed to improve the population's response (e.g. evacuation and sheltering). The urban form is assumed as an almost immutable context.

This paper aims to bridge this gap by examining, first, the role of urban form in achieving more effective and safer evacuations during near-field tsunamis; second, the paths to improve this role through appropriate urban design recommendations; and, third, the possible quantitative and qualitative tools to appraise these. To achieve this, this paper will introduce the case study of Iquique, a Chilean tsunami-prone city with current high levels of vulnerability.

The paper focuses on pedestrian evacuation, which is strongly encouraged over car-based ones in case of a near-field tsunami emergency (Imamura, 2012; IOC-UNESCO, 2008; Johnstone and Lence, 2011; Samant et al., 2008; Scheer et al., 2011; Spahn et al., 2010; Wood and Schmidtlein, 2012).

**Urban form and tsunami evacuation in Iquique, Chile**

**An earthquake and tsunami-prone city in the desert**

Iquique (20°32′S, 70°11′ W) is located 1500 km north of the Chilean capital Santiago, in the Atacama Desert from the Tarapacá region, with a population of approximately 284,000 (INE, 2012). Iquique's site is a narrow coastal plain, approximately 3 to 0.5 km wide, with its main axis (12 km long) located in a north–south direction. Its western boundary is delimited by the Pacific Ocean; its eastern edge by a 600 m high continuous steep slope. On average, the city has an elevation of roughly 25–30 m above sea level. See Figure 1.

Iquique is an earthquake and tsunami-prone territory. Lomnitz (1970) points out that, since 1535, at least five major earthquakes have had their epicentres in the northern area of Chile; in turn, at least four of these events caused major tsunamis (in 1604, 1715, 1868 and 1877). In the case of Iquique, the first of the last two major registered tsunamis struck on the 13th of August of 1868, causing more than 150 deaths and huge material losses (Donoso, 2008), with a run-up approximately 18 m above sea level (Monge and Mendoza, 1993). The second one hit the city on the 9th of May of 1877, causing around 30 deaths (Monge and Mendoza, 1993). Both were caused by near-epicentre earthquakes with an estimated moment magnitude of 8.8 (Comte and Pardo, 1991).

The last major seismic event that struck Iquique occurred on the 1st of April of 2014 at 8:46 PM, when a M8.2 earthquake hit the city with an epicentre located 85 km northwest (USGS, 2014). The earthquake brought about seven deaths, around US$100 million in material losses (Hayes et al., 2014; Labrín, 2014) and triggered a minor tsunami that hit the city's coasts at approximately 9:05 PM, with a maximum height of around 2.5 m above normal sea level. The altered ocean conditions, in turn, lasted for about 12 h (IOC, 2014). Only a small part of the northwest coast of Iquique was impacted by the waves; the rest of the city, fortunately, was not affected by the phenomenon. The population conducted a massive preventive evacuation (estimated at 60,000 people (Salinas, 2014)) after a tsunami warning was released at 8:54 PM by the Chilean Emergency Management Agency (ONEMI). Local and national authorities praised the overall success of the evacuation, although two main problems were detected: the use of cars that led to accidents and blockages in some routes, and the lack of street lighting due to the power failure provoked by the earthquake (Muñoz, 2014; Riveros, 2014).

[Figure]

**Figure 1.** Evacuation zones and total 'optimistic' evacuation times in Iquique.
Source: the authors 193×298 mm (300 × 300 DPI).

A large magnitude earthquake has been predicted for Iquique for many decades, due to the long seismic gap after the 1877 event. Scientists underline that the 1st of April of 2014 earthquake was not the expected major event and that large non-fractured seismic zones remain in the subduction junction between the Nazca and the South American plates

(Hayes et al., 2014). These segments (hundreds of kilometres long each) are located both north and south of Iquique and they could produce mega earthquakes in the future, with possible associated tsunamis (Hayes et al., 2014).

**Antecedent conditions of tsunami vulnerability in Iquique**

It can be argued that the current vulnerability to tsunamis in Iquique is the historical outcome of four interrelated antecedent conditions, also existing in other tsunami-prone contexts around the world. The first condition is the *long recurrence* of these phenomena. For a given coastal location, destructive tsunamis can have return periods of several decades or even centuries. This fact can undermine both long-term risk reduction efforts and everyday tsunami awareness among the population and authorities (especially in developing countries with multiple competing needs and limited resources), leading to urban growth patterns that increase vulnerability. For instance, between the major earthquakes of 1877 and 2014 Iquique increased its urban area from approximately 70 to 1500 ha; of these, around 770 (51.3%) are located in tsunami vulnerable zones, including extremely risky areas like the tax free trade zone (Zona Franca de Iquique, ZOFRI), with an average elevation less than 10 m (see Figure 1).

The second antecedent condition is related to *siting decisions and inertia*. Coastlines have always attracted human settlements for housing, maritime facilities and resort developments, a tendency fostered by the long gaps between devastating tsunamis (NTHMP, 2001). Once coastal towns start to develop specific location-related characteristics, 'the urbanization process tends to be definitive and in general terms irreversible' (Menoni and Pesaro, 2008: 34). Relocation, as a possible response to increased tsunami awareness, has proven difficult due to the tight bonds historically developed between the population and its site (Menoni and Pesaro, 2008; Oliver-Smith, 1991). Iquique, for instance, has historically taken advantage of its coastal location, as a small fishing cove, a mining port and now a commercial and touristic hub. Nowadays, around a 40% of the coastal population (i.e. 83,331 people) live in tsunami vulnerable zones.

A third antecedent condition is the *morphological characteristics* of the city. At a macro-scale level, certain street pattern arrangements are better suited for a rapid evacuation than others. For instance, the dense orthogonal grid is common in Latin American cities as the result of urban guidelines embodied in the 'Leyes de Indias', enacted in 1573 by King Felipe II of Spain (Wyrobisz, 1980). For evacuation purposes, the grid provides straight, redundant and nearby routes; this is an important advantage over other patterns with more 'hierarchical' layouts, like those with curvilinear distributor roads, branching patterns and many cul-de-sacs (common in contemporary urbanization) (Marshall, 2005). At a micro-scale level, in turn, certain characteristics of the urban form might seriously increase the vulnerability of evacuees. In the case of Iquique, for instance, the old central business district (CBD) is provided with narrow street sections (typically 10–11 m wide) bounded by two, three or four-storey high continuous facades; most of these, in turn, have poor seismic qualities and multiple risky elements (e.g. billboards, plasterboard decorating elements, etc.). This streetscape is also characterized by an overall poor evacuation-related design of urban furniture elements.

Finally, the fourth antecedent condition is the *actual response* of the population and emergency systems during a near-field tsunami evacuation. The analysis of past emergencies in Iquique and other tsunami-prone sites allows the identification of common harmful characteristics in this type of situations: (i) an overall collapse across the interrelated elements of human settlements (e.g. lifeline infrastructures, communication networks, etc.)

due to 'sequential failures' provoked by a large magnitude earthquake (Alesch and Siembieda, 2012; Little, 2002); (ii) little or no guidance provided by the authority (including information provision), due to sequential failures, lack of enough personnel for managing tens of thousands of evacuees and institutional needs for saving critical equipment for the post-disaster stage and (iii) evacuation-hindering behaviours among the population, such as incorrect routing, late departure, use of cars, entering vulnerable areas instead of leaving them (for instance, to pick up children from school) and returning home before the warning was cancelled.

**Research method and outcomes**

This paper is based upon a case study research method in three sequential phases: (1) a diagnosis of the existing situation, (2) a proposal of strategic urban design recommendations and (3) a critical examination of this modified scenario. In turn, each phase was developed at two different levels, each of them of crucial importance: the macro-scale of the urban tsunami evacuation system comprised of routes and safe assembly areas, and the micro-scale of the actual evacuees' experience when occupying them during an emergency. The three phases at their two scales were examined by combining quantitative and qualitative methods as described below.

*Diagnosing the existing situation.* The existing macro-scale conditions for tsunami evacuation in Iquique were examined with the aid of two computer-based models. Mas et al. (2013) point out that using tsunami evacuation models is a feasible approach due to the difficulties involved in real world 'experiments', i.e. evacuation drills: interruption of daily activities, creation of unpleasant feelings in residents and tourists, non-full population participation, difficulties for periodic repetitions, etc. The first developed model aimed to examine the urban configuration, i.e. 'the way in which the spatial elements through which people move –streets, squares, alleys, and so on– are linked together to form some kind of global pattern' (Hillier et al., 1993: 29). This configuration, in turn, determines the spatial relationship between the evacuees' vulnerable locations and safe destinations. By overlaying the Iquique tsunami flood map provided by the Chilean Navy (SHOA, 2012) and the existing urban network, a 'safety perimeter' of streets outside the vulnerable area was identified as the closest destination for evacuees (see Figure 1). Pedestrians and especially evacuees usually try to select the closest destination in terms of distance and time (Gehl, 2011; Meit et al. (2008), as cited in Mohareb, 2011; Stroehle, 2008). Drawing on this, the 'Closest Facility' function from ArcGIS's Network Analyst (which applies Dijkstra's algorithm to find, from every required location within a street network, the shortest path to one or more destinations) allowed the calculation of tens of thousands of evacuation routes, alongside the examination of their convergent spatial characteristics such as overlapping density and critical bottlenecks.

The second developed computer model, in turn, was an agent-based simulation. In this type of models, complex social systems are decomposed into units (the 'agents'), each of them following a programmed set of rules to interact between themselves and with a detailed representation of their environment (Klüpfel, 2003). The outcome of this process, after a certain number of iterations in a timeline, is called complex, emergent or collective behaviour, which in the case of evacuation analyses allows to calculate the required time to remove all the agents from an endangered area (Chen and Zhan, 2008; Lämmel et al., 2010). The agent-based model was developed in Agent Analyst, an ArcGIS plugin (http://resources.arcgis.com/en/help/agent-analyst/) that integrates Repast (a popular agent-based

**Table 1.** Terrain slope and pedestrian speed conservation.

| Slope (degrees) | Speed conservation (%) |
| --- | --- |
| 0 | 100 |
| 0–5 | 90 |
| 5–15 | 80 |
| 15–30 | 40 |
| 30–45 | 15 |
| More than 45 | 5 |

Source: Adapted from Post et al. (2009).

modelling platform) with the ArcGIS environment (Johnston, 2013) While this integration allows the direct definition of the model's main elements (e.g. agents and environment) from ArcGIS shapefiles, the interaction conditions need to be programmed in the ad hoc language 'Not Quite Python' (Ligmann-Zielinska, 2013).

In the model, every evacuee in the tsunami vulnerable area (i.e. an 'agent') was provided with a shortest route to follow (obtained from the previous network model), a base pedestrian evacuation speed (1.4 m/s) (Ando et al. (1988), as cited in Smith, 1995), and three speed-decreasing rules, which were combined with equal weights and applied by every agent during each step of the model. Two of the rules were related to the interaction with the environment. First, steeper street slopes (collected from a digital elevation model) lead to decreased walking speeds, according to the parameters proposed by Post et al. (2009) (see Table 1). Second, existing traffic conditions and streets' characteristics (collected from sources such as satellite images, maps and Google Street View) can have a direct impact on pedestrian speed by reducing the available street space for evacuation, as pointed out by Mück (2008) (see Table 2). The third speed-decreasing rule, in turn, was related to the spatial interaction between agents as they moved in the model; this, because larger densities of pedestrians lead to diminished walking speeds, as suggested by Ando et al. (1988) (as cited in Smith, 1995) (see Figure 2). As the main purpose of the agent-based model was to examine the impact of the urban form (and its changes) on the evacuation process in Iquique, other dynamic conditions (like the evacuees' ability to re-route in case of emerging congestions or blockages in the network) were not included in this analysis. See snapshots of the model in Figure 3.

The urban configuration and the agent-based models required as inputs the spatial distributions of evacuees for both daytime and night-time examined scenarios. These were obtained from an origin–destination study conducted in Iquique in 2010 by the Chilean Ministry of Transport (SECTRA, 2014), with total vulnerable populations of 108,881 and 83,331 during daytime and night-time, respectively. For the purpose of reducing computer memory and time consumptions, in turn, the city was divided into five evacuation zones that were examined separately with the models, following the approach proposed by Imamura et al. (2012). These zones were defined according to the existing Iquique Municipality's City Response Plan to an Emergency or Disaster (Iquique, 2011) (see Figure 1). Finally, in the case of the agent-based model, two different evacuees' departure scenarios were tested: a 'full compliance' one where all the agents started their evacuation at the same time (i.e. an 'optimistic' model, which in the real world could be triggered by the perception of the earthquake itself and a high-penetration warning system) and a 'delayed' scenario where few evacuees departed at the beginning of the simulation, then the number increased and then slowly decreased until it stopped. This 'pessimistic' scenario (more similar to a real

**Table 2.** Estimated pedestrian speed conservation factors.

| Road classification | | | | | |
| --- | --- | --- | --- | --- | --- |
| Width_type | 1 | 1 | 1 | 2 | 2 |
| Traffic volume_type | 1 | 2 | 3 | 1 | 2 |
| Speed conservation (%) | 50% | 65% | 80% | 55% | 70% |
| Width_type | 2 | 3 | 3 | 3 | 4 |
| Traffic volume_type | 3 | 1 | 2 | 3 | 1 |
| Speed conservation (%) | 85% | 60% | 75% | 90% | 75% |
| Width_type | 4 | 4 | 5 | 5 | 5 |
| Traffic volume_type | 2 | 3 | 1 | 2 | 3 |
| Speed conservation (%) | 80% | 95% | 85% | 90% | 100% |
| Legend | | | | | |
| Width | 1 | One lane small | | | |
| | 2 | One lane wide | | | |
| | 3 | Two lanes small | | | |
| | 4 | Two lanes wide | | | |
| | 5 | Four lanes | | | |
| Traffic volume | 1 | High | | | |
| | 2 | Medium | | | |
| | 3 | Low | | | |

Source: Adapted from Mück (2008).

[Figure]

**Figure 2.** Uni-directional walking speed as a function of pedestrians density.
Source: Adapted from Ando et al. (1988), as cited in Smith (1995) 192×147 mm (300 × 300 DPI).

emergency) was modelled according to the agents randomly selecting their evacuation start times from a Rayleigh cumulative distribution (Mas et al., 2013), with an average departure time of 20 min after the beginning of the evacuation, i.e. the expected arrival time of a near-field tsunami (Walker, 2013). For this stochastic scenario, 50 simulation rounds were conducted to obtain an average value of the required times of evacuation for every zone. This number of rounds was sufficient to achieve, for every zone's average total evacuation

[Figure]

**Figure 3.** Snapshots from the Iquique agent-based evacuation model, showing the evacuees (dark points), the urban network (dark lines), the assembly points (circles) and the expected flooding area (grey line). Source: the authors 299 × 150 mm (300 × 300 DPI).

time, a 95% confidence interval with a margin of error less than 1 min (see Figures 1, 4 and 5).

The existing micro-scale conditions, in turn, were examined during a fieldwork study conducted in April 2013. This analysis included a thorough survey of existing risk factors along the priority evacuation routes, i.e. those streets that might gather the largest numbers of escaping paths during an emergency. The priority routes were identified in the urban configuration model by applying to the evacuees' found paths the 'Line Density' function from the ArcGIS' Spatial Analyst toolkit. It was found that 21 routes, adequately distributed from north to south in Iquique, gathered around 55% of all the evacuees' paths (see Figure 6). The risk factors, in turn, were identified through an extensive literature review on urban safety parameters (Ciborowski, 1982; Davidson and Shah, 1997; Ercolano, 2008; Erdik, 1994; He and Xu, 2012; Nadel, 2004; NFPA, 2012; NYSDOT, 2013; Scheer et al., 2012), including two tsunami evacuation drills conducted in Chile in 2013.

*Diagnosis' outcomes.* A near-field tsunami in Iquique is expected to arrive to the city's coasts in around 20 min after the related earthquake. The results of the diagnosis phase at the macro-scale (summarized in Figures 1 and 5, Tables 3 to 5) show that in the case of an 'optimistic' evacuation scenario (i.e. with non-delayed departures) up to 90% of the population during both daytime and night-time emergencies could reach a safe destination within this time. In the case of a 'pessimistic' scenario, in turn, only 26% of the daytime and 30% of the night-time populations could reach a safe area within this time. In turn, a complete evacuation of the city could take around one and a half hours.

Despite the overall positive characteristics of an 'optimistic' evacuation scenario, the findings show that some parts of Iquique remain significantly vulnerable, as their inhabitants would not be able to complete an evacuation in case of a near-field tsunami emergency. Particularly, the ZOFRI located in zone 1, the port site in zone 2 and the

[Figure]

**Figure 4.** Rayleigh cumulative distribution of probable evacuation times, with an average value of 20 min.
Source: Adapted from Mas et al. (2013) 229 × 170 mm (300 × 300 DPI).

[Figure]

**Figure 5.** Total number of safe evacuees in time, Iquique, for existing/modified, daytime/night-time and 'optimistic'/'pessimistic' evacuation scenarios.
Source: the authors 299 × 192 mm (300 × 300 DPI).

[Figure]

**Figure 6.** Synthesis of suggested macro-scale urban changes in Iquique.
Source: the authors 297 × 165 mm (300 × 300 DPI).

Cavancha Peninsula and Beach in zones 3 and 4 have evacuation times that can extend up to 42, 36, 33 and 28 min, respectively, even during an 'optimistic' scenario (see Figure 1).

At the micro-scale level, in turn, the fieldwork analysis of the previously identified 21 priority evacuation routes allowed the identification of 393 vulnerable points along approximately 16.87 km of streets, i.e. around one every 43 m. These issues can be classified into three main categories. First, poor physical conditions and inadequate maintenance of the urban environment, including: critical street sections where seismically vulnerable high facades are located too close to the sidewalk; and damaged or absent pavements on sidewalk and roadways. Second, issues derived from a poor

**Table 3.** Comparison between existing and modified scenarios, urban configuration model, Iquique.

Iquique urban configuration analysis

| Zone | Population | | Average distance to the safe assembly area (km) | | Longest evacuation route (km) | | | | Average evacuation route (km) | | | |
| | Daytime | Night-time | Daytime | Night-time | Existing | | Modified | | Existing | | Modified | |
| | | | | | Daytime | Night-time | Daytime | Night-time | Daytime | Night-time | Daytime | Night-time |
|---|---|---|---|---|---|---|---|---|---|---|---|---|
| 1 | 16,840 | 14,352 | 0.43 | 0.25 | 2.63 | 2.63 | 1.74 | 1.14 | 0.64 | 0.39 | 0.60 | 0.39 |
| 2 | 34,543 | 20,483 | 0.55 | 0.49 | 2.69 | 2.42 | 1.50 | 1.50 | 0.64 | 0.57 | 0.61 | 0.57 |
| 3 | 31,015 | 25,873 | 0.42 | 0.31 | 1.924 | 1.924 | 1.63 | 1.63 | 0.55 | 0.41 | 0.53 | 0.40 |
| 4 | 22,556 | 18,960 | 0.46 | 0.45 | 2.06 | 2.06 | 2.06 | 2.06 | 0.62 | 0.59 | 0.61 | 0.59 |
| 5 | 3,927 | 3,663 | 0.24 | 0.22 | 1.07 | 1.06 | N/A | N/A | 0.48 | 0.44 | N/A | N/A |

N/A means that no macro-scale changes were suggested.
Source: the authors.

**Table 4.** Total evacuation times for existing and modified 'optimistic' scenarios, agent-based model, Iquique.

Iquique existing/modified and daytime/night-time 'optimistic' evacuation scenarios

| Zone | Main land use | Scenario | Population | Total evacuation time (min) | |
|------|---------------|----------|-----------|----------|----------|
| | | | | Existing | Modified |
| 1 | ZOFRI + residential | Daytime | 16,840 | 42 | 28 |
| | | Night-time | 14,352 | 42 | 19 |
| 2 | Port + CBD | Daytime | 34,543 | 36 | 23 |
| | | Night-time | 20,483 | 33 | 23 |
| 3 | Residential + services + Cavancha Beach | Daytime | 31,015 | 28 | 22 |
| | | Night-time | 25,873 | 25 | 21 |
| 4 | Residential + services | Daytime | 22,556 | 33 | 33 |
| | | Night-time | 18,960 | 33 | 33 |
| 5 | Residential | Daytime | 3,927 | 17 | N/A |
| | | Night-time | 3,663 | 17 | N/A |

N/A means that no macro-scale changes were suggested.
Source: the authors.

**Table 5.** Average total evacuation times for existing and modified 'pessimistic' scenarios, agent-based model, Iquique.

Iquique existing/modified and daytime/night-time 'pessimistic' evacuation scenarios

| Zone | Main land use | Scenario | Population | Total evacuation time (min) | |
|------|---------------|----------|-----------|----------|----------|
| | | | | Existing | Modified |
| 1 | ZOFRI + residential | Daytime | 16,840 | 88.1 | 79.2 |
| | | Night-time | 14,352 | 69.8 | 68.3 |
| 2 | Port + CBD | Daytime | 34,543 | 81.8 | 75.6 |
| | | Night-time | 20,483 | 79.2 | 75.5 |
| 3 | Residential + services + Cavancha Beach | Daytime | 31,015 | 80.8 | 74.9 |
| | | Night-time | 25,873 | 76.9 | 72.4 |
| 4 | Residential + services | Daytime | 22,556 | 84.9 | 84.7 |
| | | Night-time | 18,960 | 85.6 | 85.6 |
| 5 | Residential | Daytime | 3,927 | 68.5 | N/A |
| | | Night-time | 3,663 | 68.1 | N/A |

N/A means that no macro-scale changes were suggested.
Source: the authors.

evacuation-related design of the public space, including: inadequate sidewalk characteristics (including steps, sudden narrowing or end of the footway, impeding elements such as wrongly placed fences, kiosks, bollards, flower beds, agglomeration of trees, etc.); an existing dense above-ground network for electricity distribution (including wires, poles and transformers); hazardous elements that are external to the pedestrian space, such as hanging billboards, poor-quality cantilever facades and balconies, and gas tanks; and lack of support systems for emergency conditions (e.g. electrical power blackout in Iquique is

common following a large magnitude earthquake). The third type of issue for evacuation at the micro-scale level is related to inappropriate evacuation-related uses of the public space, which include: high levels of vehicular traffic, cars commonly parked on the sidewalk and occupation of the sidewalk space with transitory activities such as restaurant tables and street trading.

*Urban form changes for tsunami evacuation.* The diagnosis phase allowed the identification of critical parts of the urban realm where feasible interventions could be developed to improve existing conditions for tsunami evacuation; this is a type of 'what-if' scenario analysis, which investigates 'what will happen on the condition of some specified near-future events of great importance for future development' (Börjeson et al., 2006: 726). At the macro-scale, interventions were developed for the three areas with the highest total evacuation times in Iquique, as found during the diagnosis phase (see Figures 1 and 6): the ZOFRI (zone 1), the port site (zone 2) and the Cavancha Beach (zone 3). In the first two cases, the proposal includes the creation of vertical evacuation points (e.g. tsunami-resistant towers), whilst in the Cavancha Beach it is suggested the provision of a new pedestrian connection across the former airport lot. The densely built Cavancha Peninsula, in turn, could use some of its existing high-rise buildings as public vertical evacuation points. This option, however, has been deemed unfeasible by local emergency planners due to the inability of guaranteeing the stability of these buildings in front of the tsunami's distribution of forces (Araneda, 2014; Gallardo, 2014) and therefore is not part of this analysis. A final macro-scale change is suggested for the densely populated CBD: the pedestrianization of approximately 800 m in four of its narrow streets, to mitigate the threat to evacuees currently posed by the excess of vehicular traffic (see Figure 6).

At the micro-scale level, in turn, the modification scheme aimed to improve the evacuation-related characteristics of the 21 priority routes identified in Iquique. The proposal aims to mitigate the vulnerabilities identified during the diagnosis phase with an appropriate urban design, especially in the most critical parts of the routes, like those located adjacent to the coast and in the CBD. Safety and wayfinding systems such as solar-powered street lighting, continuous pavements for guidance and tsunami signage should also be incorporated. By combining micro-scale with macro-scale modifications (e.g. pedestrianization), in turn, an overall enhancement of the urban quality could be achieved (see Figure 7).

Finally, micro-scale modifications should also aim to generate a network of small–medium assembly spaces along the aforementioned 'safety perimeter' for sheltering the evacuees during several hours. Currently, Iquique is lacking open public spaces outside the floodable area capable of providing shelter and supporting the delivery of first aid and other disaster-related services. Recent tsunami emergencies in the city were characterized by disorderly gatherings of evacuees in remnant or non-built urban areas, sometimes not clearly separated from vehicular traffic and lacking any kind of infrastructure or amenities.

*Assessment of the modified situation.* The proposed macro-scale modifications in Iquique were examined with the aid of the urban configuration and agent-based models introduced above. The changes in the evacuation routes, the total required times for evacuation and the rate of safe evacuees in time are summarized in Tables 3 to 5 and Figure 5.

The urban configuration model shows the importance of an appropriate street pattern in providing expedited connections between vulnerable and safe urban areas. The evacuees' initial Euclidean distances to the safe destinations have a direct impact on the overall length of the evacuation routes; this intrinsic vulnerability can be mitigated by an appropriately

[Figure]

**Figure 7.** Reference image of micro-scale changes suggested for Tarapacá Street, Iquique.
Source: the authors 248 × 299 mm (300 × 300 DPI).

dense and connected urban grid. On this point, Dill (2004: 4) suggests the Pedestrian Route Directness (PRD) index as 'the ratio of route distance to straight-line distance for two selected points'. For instance, in Iquique zone 2 has the lowest average PRD (1.16 for both daytime and night-time emergencies), whilst zone 5 has the highest one (2.0 in both scenarios). Zone 2 is a much more 'connected' area than zone 5: it is provided with a street density of 18.76 (km/km$^2$) and an intersection density of 127.03 (junctions/km$^2$), versus 11.12 (km/km$^2$) and 78.01 (junctions/km$^2$), respectively, in the case of zone 5. As the result of these conditions, evacuation routes in zone 5 are proportionally larger than those in zone 2 and include more changes of direction during the trip between vulnerable and safe areas, which might imply wayfinding (i.e. 'the process of determining and following a *path* or *route* between an origin and a destination' (Golledge, 1999: 6)) issues for evacuees.

The urban configuration model also shows that the total length of the evacuation routes could be significantly reduced in the zones where the proposal includes new vertical evacuation points, i.e. zone 1 and 2 (see Figures 1 and 6): a 33.84 and 44.24% of reduction during the daytime scenarios, and a 56.65 and a 38.01% during the night-time emergencies, respectively. In the case of the new pedestrian crossing proposed in zone 3, where the safe destination remains the same, the impact of the reduction is less, 15.28%. The reduction on the average evacuation routes, in turn, is also determined by the likely ratio of evacuees using the new facilities. The proposed vertical evacuation points are expected to be built and operated by the ZOFRI and the Port and located in sites owned by these companies. Therefore, they would have a limited number of potential users (i.e. employees), mainly during a daytime scenario: 479 (ZOFRI) and 620 evacuees (port), i.e. only around a 2.84 and a 1.79% of the total daytime populations in zone 1 and 2, respectively. The related average length of evacuation routes, in turn, is reduced by 6.25% (zone 1) and 4.69% (zone 2), whilst the night-time scenarios are not affected. In zone 3, around 5000 evacuees, located mainly at the Cavancha Beach (i.e. a 16.1% of the zone's population) are the users of the new pedestrian crossing. In this case, a reduction of a 1.61%

in the average length of the daytime evacuation routes is expected (with no changes in the night-time scenario, when the beach is practically empty).

The agent-based model, in turn, shows that the reduction in the largest routes' lengths could lead to noticeable decreases in the total required evacuation times: up to a 54.76% (zone 1, night-time) in an 'optimistic' scenario, and up to a 10.1% (zone 1, daytime) during a 'pessimistic' one. The improvement in rate of safe evacuees in time, in turn, is limited by the number of 'agents' that actually use the proposed facilities in the model. In the case of the 'optimistic' scenario, the largest improvement is achieved at around the 20 min mark of evacuation, with an 'extra' number of 4629 and 1827 evacuees during the daytime and night-time emergencies, respectively. The 'pessimistic' scenario, in turn, exhibits its greatest change (although a minor one, when compared to the 'optimistic' model) at minute 35 of evacuation, with 2386 and 782 'extra' evacuees during the daytime and night-time modified scenarios, respectively.

**Discussion**

Iquique's significant vulnerability to near-field tsunamis is the historical outcome of antecedent conditions that include the long recurrence between disasters, the need to take advantage of coastal proximity, urban development patterns and the actual responses of the exposed populations during emergencies. Usual long-term efforts to mitigate this vulnerability, like civil-engineered defences or land use changes, can be difficult or unfeasible to implement in a developing country like Chile, with multiple competing needs and limited resources. It seems appropriate, then, to strategically focus efforts on evacuation, 'the most important and effective method to save human lives' during a tsunami (Shuto, 2005: 8). Particularly, this paper examined how macro- and micro-scale urban form changes related to street patterns and safe assembly areas in Iquique can lead to significant reductions on the required times for evacuation and to safer conditions for evacuees. It was also suggested that these changes can trigger an overall improvement of the urban realm and provide the population with enhanced open public spaces, to be enjoyed by the population especially during non-emergency times.

These changes were proposed to mitigate the most urgent evacuation-related issues detected during the diagnosis phase, including a 'feasibility' check provided by fieldwork; they need to be understood as a necessary first step of an iterative improvement strategy. There are multiple possible interventions that could be tested in Iquique. Although it is beyond the scope of this paper, specific methods could be applied to compare multiple possible interventions and identify one or more preferred solutions; see for instance a 'multi-criteria decision analysis' for tsunami vertical-evacuation refuges in Wood et al. (2014) and multiple scenario testing in Di Mauro et al. (2013).

It is also important to underline the large existing differences between the 'optimistic' and 'pessimistic' scenarios examined in this paper, as the result of rapid versus delayed evacuations. To this respect, it has been argued that the populations' decision to evacuate is complex to predict and relates to risk perception, social factors and access to information (Imamura et al., 2012; Strunz et al., 2011). Nevertheless, this paper suggests that ongoing risk-reduction efforts in Iquique point in the right direction by fostering evacuation responses that might be closer to an 'optimistic' scenario in case of an emergency. These efforts include the installation in 2013 of 17 long-range klaxon alarms for warning spreading, and educational campaigns based on information provision (via TV, radio and brochures) and five large tsunami evacuation drills conducted in the city in 2008, 2010, 2011, 2012 and 2013. The suggested micro-scale changes could also contribute to an 'ongoing evacuation

education' in Iquique, by the incorporation of proper signage or other related changes (for instance, renaming streets or public spaces according to its evacuation role).

In this paper, micro-scale vulnerabilities were examined during the fieldwork phase, leading to the suggestion of qualitative changes for priority evacuation routes as those shown in Figure 7. Future research in this area could benefit from larger surveys to provide a more quantitative analysis. This could be done, for instance, by developing several 'failure scenarios' where the spatial agglomeration of micro-scale features might lead to the identification of possible bottlenecks or blockage points during an emergency. These points, in turn, could be loaded into an enhanced agent-based model including more dynamic characteristics, like the capacity of re-routing in front of those critical conditions when they are 'activated' as model parameters. Another possibility is to convert the surveyed micro-vulnerabilities into 'cost surfaces' for least-cost-distance analysis (see for instance Laghi and Cavalletti (2006)) to find the shortest paths between vulnerable and safe locations in the macro-scale analysis.

**Conclusion**

This paper has argued that urban form can have an essential role in responding to rapid onset disasters, particularly by achieving safer and more effective evacuations in case of near-field tsunamis. Two different scales were examined in the case study of Iquique, Chile: the macro-scale of the urban configuration (including street pattern arrangements and location of safe areas) and the micro-scale of the built environment conditions experienced by the evacuees during a possible evacuation. The paper suggested a mixed methods approach (computer-based models and fieldwork) developed in three sequential phases including the diagnosis of the existing situation, the proposal of strategic urban design recommendations for improvement and a critical analysis of the modified scenario.

The results showed significant existing evacuation vulnerability in Iquique at both scales of analysis, when even during 'optimistic' scenarios a total evacuation cannot be completed in the expected arrival time of a near-field tsunami (20 min). The developed models also showed that the suggested urban modifications, in turn, are capable of improving the existing conditions especially by reducing the total evacuation times in some critical areas. The rate of safe evacuees in time could also be enhanced, but the relative importance of this improvement decreases as the evacuees' departure time moves from an 'optimistic' to a 'pessimistic' scenario. Moreover, the overall impact of the design recommendations is limited by the actual number of expected users. By developing further urban interventions under an iterative strategy of multiple-scenarios testing, greater improvements could potentially be achieved.

It is necessary to underline that this paper examined theoretical responses to an urban design and disaster vulnerability problem. For their implementation in a 'real world' context, urban modifications as those suggested should undergo a complex process of testing and validation against multiple constraints and competing needs. This process might include: gathering of political, social and financial support; analysis of environmental, real estate and transit implications; and study of engineering requirements, among others.

**Declaration of conflicting interests**

The author(s) declared no potential conflicts of interest with respect to the research, authorship, and/or publication of this article.

**Funding**

The author(s) received no financial support for the research, authorship, and/or publication of this article.

**References**

Alesch DJ and Siembieda W (2012) The role of the built environment in the recovery of cities and communities from extreme events. *International Journal of Mass Emergencies and Disasters* 30: 197–211.

Allan P and Bryant M (2010) *The Critical Role of Open Space in Earthquake Recovery: A Case Study*. Wellington: New Zealand Society of Earthquake Engineering Conference.

Allan P, Bryant M, Wirsching C, et al. (2013) The influence of urban morphology on the resilience of cities following an earthquake. *Journal of Urban Design* 18: 242–262.

Allan P and Roberts J (2009) Urban resilience and the open space network. *Tephra* 22: 55–59.

Araneda Y (2014) Autoridades y expertos enfatizan en evacuación vertical en edificios. *La Estrella de Iquique*, 19 March 2014, 3.

Bernard EN (1995) *Tsunami Hazard Mitigation: A Report to the Senate Appropriations Committee*. Seattle: National Oceanic and Atmospheric Administration.

Börjeson L, Höjer M, Dreborg K-H, et al. (2006) Scenario types and techniques: Towards a user's guide. *Futures* 38: 723–739.

Burby R, Beatley T, Berke P, et al. (1999) Unleashing the power of planning to create disaster-resistant communities. *Journal of the American Planning Association* 65: 247–258.

Cai K and Wang J (2009) Urban design based on public safety—Discussion on safety-based urban design. *Frontiers of Architecture and Civil Engineering in China* 3: 219–227.

Carmona M, Tiesdell S, Heath T, et al. (2010) *Public Places, Urban Spaces. The Dimensions of Urban Design*. Oxford: Elsevier.

Chen X and Zhan FB (2008) Agent-based modelling and simulation of urban evacuation: Relative effectiveness of simultaneous and staged evacuation strategies. *Journal of the Operational Research Society* 59: 25–33.

Ciborowski A (1982) Physical development planning and urban design in earthquake-prone areas. *Engineering Structures* 4: 153–160.

Comte D and Pardo M (1991) Reappraisal of great historical earthquakes in the Northern Chile and Southern Peru seismic gaps. *Natural Hazards* 4: 23–44.

Conzen MRG (1960) Alnwick, Northumberland: A study in town-plan analysis. *Transactions and Papers (Institute of British Geographers)* 27: iii–122.

Cutter S, Barnes L, Berry M, et al. (2008) A place-based model for understanding community resilience to natural disasters. *Global Environmental Change* 18: 598–606.

Database ID (2011) Disaster trends. Available at: http://www.emdat.be/disaster-trends (accessed 20 March 2012).

Davidson RA and Shah HC (1997) *An Urban Earthquake Disaster Risk Index*. Stanford: Department of Civil and Environmental Engineering, Stanford University.

Dill J (2004) Measuring network connectivity for bicycling and walking. In: *83rd Annual Meeting of the Transportation Research Board*, Washington DC (CD-ROM): Transportation Research Board.

Di Mauro M, Megawati K, Cedillos V, et al. (2013) Tsunami risk reduction for densely populated Southeast Asian cities: Analysis of vehicular and pedestrian evacuation for the city of Padang, Indonesia, and assessment of interventions. *Natural Hazards* 68: 373–404.

Donoso C (2008) 1868: Un annus horribilis en la historia de Iquique. *Revista de Ciencias Sociales, Universidad Arturo Prat* 20: 37–60.

Ercolano JM (2008) Pedestrian disaster preparedness and emergency management of mass evacuations on foot: State-of-the-art and best practices. *Journal of Applied Security Research* 3: 389–405.

Erdik M (1994) Developing a comprehensive earthquake disaster master plan for Istanbul. In: Tucker BE, Erdik M and Hwang CN (eds) *Issues in Urban Earthquake Risk*. Dordrecht: Springer Science+Business Media, pp. 125–166.

Gallardo K (2014) Iquique se abre al debate sobre la evacuación vertical. Available at: http://www.24horas.cl/nacional/iquique-se-abre-al-debate-sobre-la-evacuacion-vertical-1149457 (accessed 29 March 2014).

Gehl J (2011) *Life Between Buildings: Using Public Space*. Washington DC, USA: Island Press.

Golledge R (1999) Human wayfinding and cognitive maps. In: Golledge R (ed.) *Wayfinding Behavior*. Baltimore: The John Hopkins University Press, pp. 5–45.

Hayes GP, Herman MW, Barnhart WD, et al. (2014) Continuing megathrust earthquake potential in Chile after the 2014 Iquique earthquake. *Nature* 512: 295–298.

He L and Xu S (2012) Urban design strategies of public space based on danger stress response. *Advanced Materials Research* 450: 1026–1031.

Hillier B, Penn A, Hanson J, et al. (1993) Natural movement: Or, configuration and attraction in urban pedestrian movement. *Environment and Planning B: Planning and Design* 20: 29–66.

Imamura F (2012) *Lessons from a Disaster: The Great East Japan Earthquake Recorded on Camera*. Sendai Television: Sendai.

Imamura F, Muhari A, Mas E, et al. (2012) Tsunami disaster mitigation by integrating comprehensive countermeasures in Padang City, Indonesia. *Journal of Disaster Research* 7: 48–64.

INE (2012) *Programa de Proyecciones de la Población*. Santiago: Gobierno de Chile.

IOC (2014) Sea level station monitoring facility. Available at: http://www.ioc-sealevelmonitoring.org/station.php?code=iqui2 (accessed 4 April 2014).

IOC-UNESCO (2008) Tsunami preparedness–information guide for disaster planners. In: *IOC Manuals and Guides*. Paris: UNESCO.

Iquique MD (2011) *Plan de Respuesta Comunal frente a situación de emergencia y/o desastre*. Municipalidad de Iquique: Iquique.

Ishikawa M (2002) Landscape planning for a safe city. *Annals of Geophysics* 45: 833–841.

Joerin J and Shaw R (2010) Climate change adaptation and urban risk management. In: Shaw R, Pulhin J and Pereira J (eds) *Climate Change Adaptation and Disaster Risk Reduction: Issues and Challenges (Community, Environment and Disaster Risk Management)*. Bingley: Emerald, pp. 195–215.

Johnston KM (2013) An introduction to agent-based modeling. In: Johnston KM (ed.) *Agent Analyst. Agent-Based Modeling in ArcGIS*. Redlands: Esri PRESS, pp. 1–30.

Johnstone WM and Lence BJ (2011) Use of flood, loss and evacuation models to assess exposure and improve a community tsunami response plan: A Vancouver Island case study. *Natural Hazards Review* 13: 162–171.

Kazusa S (2004) *Tsunami and Storm Surge Hazard Map Manual*. Tokyo: COD Management.

Klüpfel H (2003) *A Cellular Automaton Model for Crowd Movement and Egress Simulation*. Duisburg: Science Duisburg–Essen.

Labrín S (2014) Iquique: el día después del terremoto. *La Tercera*, 3 April 2014, 14–15.

Laghi M and Cavalletti A (2006) *Evacuation Routes Tools ArcGIS Toolbox*. User's manual.

Lämmel G, Rieser M and Nagel K (2010) Large scale microscopic evacuation simulation. In: Klingsch W, Rogsch C, Schadschneider A, et al. (eds) *Pedestrian and Evacuation Dynamics 2008*. Berlin: Springer-Verlag.

Ligmann-Zielinska A (2013) Not quite python. In: Johnston KM (ed.) *Agent Analyst. Agent-Based Modeling in ArcGIS*. Redlands: Esri Press, pp. 525–542.

Little RG (2002) Controlling cascading failure: Understanding the vulnerabilities of interconnected infrastructures. *Journal of Urban Technology* 9: 109–123.

Lomnitz C (1970) Major earthquakes and tsunamis in Chile during the period 1535 to 1955. *International Journal of Earth Sciences* 59: 938–960.

Marshall S (2005) *Streets and Patterns*. New York: Spon Press.

Mas E, Adriano B and Koshimura S (2013) An integrated simulation of tsunami hazard and human evacuation in La Punta, Peru. *Journal of Disaster Research* 8: 285–295.

Menoni S and Pesaro G (2008) Is relocation a good answer to prevent risk?: Criteria to help decision makers choose candidates for relocation in areas exposed to high hydrogeological hazards. *Disaster Prevention and Management* 17: 33–53.

Mohareb NI (2011) Emergency evacuation model: Accessibility as a starting point. *Proceedings of the Institution of Civil Engineers. Urban Design and Planning* 164: 215–224.

Monge J and Mendoza J (1993) Study of the effects of tsunami on the coastal cities of the region of Tarapacá, north Chile. *Tectonophysics* 218: 237–246.

Mück M (2008) *Tsunami Evacuation Modelling. Development and Application of a Spatial Information System Supporting Tsunami Evacuation Planning in South-West Bali*. Regensburg: Institut für Geographie Universität Regensburg.

Muñoz D (2014) Más de 120.000 personas fueron evacuadas en Arica e Iquique tras sismo. *La Tercera*, 2 April 2014, 3.

Murata S, Imamura F, Katoh K, et al. (2010) *Tsunami. To Survive from Tsunami*. Singapore: World Scientific Publishing Co. Pte. Ltd.

Nadel BA (2004)   Home and business security, disaster planning, response, and recovery. In: Nadel BA (ed.) *Building Security: Handbook for Architectural Planning and Design*. New York: McGraw-Hill.

NFPA (2012) *NFPA 101: Life Safety Code*. Quincy: National Fire Protection Association.

NTHMP (2001) *Designing for Tsunamis. Seven Principles for Planning and Designing for Tsunami Hazards*. Sacramento, CA: NTHMP.

NYSDOT (2013) Pedestrian facility design. In: *Highway Design Manual n.71*. New York: New York State Department of Transportation.

Oliver-Smith A (1991) Successes and failures in post-disaster resettlement. *Disasters* 15: 12–23.

Pearce L (2003) Disaster management and community planning, and public participation: How to achieve sustainable hazard mitigation. *Natural Hazards* 28: 211–228.

Post J, Wegscheider S, Mück M, et al. (2009) Assessment of human immediate response capability related to tsunami threats in Indonesia at a sub-national scale. *Natural Hazards and Earth System Sciences* 9: 1075–1086.

Preuss J (1988) *Planning for Risk: Comprehensive Planning for Tsunami Hazard Areas*. Arlington, VA: Urban Regional Research, National Science Foundation.

Riveros J (2014) Automovilistas iquiqueños no dejaban huir a pie. *Las Últimas Noticias*, 2 April 2014, 14.

Salinas E (2014) Cinco muertos dejó el sismo y alerta de tsunami en Iquique. *La Estrella de Iquique*, 2 April 2014, 3.

Samant LD, Tobin LT and Tucker B (2008) *Preparing Your Community for Tsunamis: A Guidebook for Local Advocates*. Palo Alto: GeoHazards International.

Scheer S, Gardi A, Guillande R, et al. (2011) *Handbook of Tsunami Evacuation Planning*. Luxembourg: Publications Office of the European Union.

Scheer S, Varela V and Eftychidis G (2012) A generic framework for tsunami evacuation planning. *Physics and Chemistry of the Earth* 49: 79–91.

SECTRA (2014) Encuestas de Movilidad de Centros Urbanos. Available at: http://sintia.sectra.gob.cl/ (accessed 12 January 2014).

SHOA (2012) *Iquique: Carta de Inundación Por Tsunami*. Valparaíso: SHOA.

Shuto N (2005) Tsunamis: Their coastal effects and defense works. In: Tingsanchali T (ed.) *Scientific Forum on the Tsunami, its Impact and Recovery*. Thailand: Asian Institute of Technology, pp. 1–12.

Shuto N and Fujima K (2009) A short history of tsunami research and countermeasures in Japan. *Proceedings of the Japan Academy, Series B* 85: 267–275.

Smith RA (1995) Density, velocity and flow relationships for closely packed crowds. *Safety Science* 18: 321–327.

Spahn H, Hoppe M, Usdianto B, et al. (2010) *Planning for Tsunami Evacuations: A Guidebook for Local Authorities and other Stakeholders in Indonesian Communities*. Jakarta: GITEWS.

Stroehle J (2008) *How Do Pedestrian Crowds React When They Are in an Emergency Situation-Models and Softwares*. Urbana-Champaign, IL: University of Illinois at Urbana-Champaign.

Strunz G, Post J, Zosseder K, et al. (2011) Tsunami risk assessment in Indonesia. *Natural Hazards and Earth System Science* 11: 67–82.

Tarrant M (2006) Risk and emergency management. *The Australian Journal of Emergency Management* 21: 9–14.

UNISDR (2004) *Living with Risk. A Global Review of Disaster Reduction Initiatives*. Geneva: UNISDR.

USGS (2014) M8.2–95 km NW of Iquique, Chile. Available at: http://earthquake.usgs.gov/earthquakes/eventpage/usc000nzvd#summary (accessed 8 April 2014).

Walker J-M (2013) Informe Técnico de Evaluación. Simulacro Macrozona de Terremoto y Tsunami, Evacuación del Borde Costero. Regiones de Arica y Parinacota, Tarapacá, Antofagasta y Atacama. 8 de Agosto de 2013, Ed ONEMI (Santiago).

Wamsler C (2014) *Cities, Disaster Risk and Adaptation*. New York: Routledge.

Wood N, Jones J, Schelling J, et al. (2014) Tsunami vertical-evacuation planning in the US Pacific Northwest as a geospatial, multi-criteria decision problem. *International Journal of Disaster Risk Reduction* 9: 68–83.

Wood NJ and Schmidtlein MC (2012) Anistropic path modeling to assess pedestrian-evacuation potential from Cascadia-related tsunamis in the US Pacific Northwest. *Natural Hazards* 62: 275–300.

Wu Y (2012) Urban spatial system planning of disaster prevention and refuge. *Advanced Materials Research* 450: 1061–1064.

Wyrobisz A (1980) La ordenanza de Felipe II del año 1573 y la construcción de ciudades coloniales españolas en la América. *Estudios Latinoamericanos* 7: 11–34.

**Jorge León** is a PhD candidate at the Faculty of Architecture, Building and Planning from the University of Melbourne (Australia), and an assistant professor at the Department of Architecture from the Universidad Técnica Federico Santa María (Chile).

**Alan March** is Associate Professor at the Faculty of Architecture, Building and Planning from the University of Melbourne (Australia).

---

## Referee Comment (RC2) · Dr Rahayu (Referee) · 13 Apr 2018

In overall, the idea is very interesting and innovative in assessing and analyzing micro scale vulnerability which needed to mitigate in order to enhance designed capacity of evacuation planning. This will be very useful for developing countries with similar situation. The data and analysis is well presented, however the approach and methodology is not clear to achieve the objective of the paper. This will be good if presented in a clear graphical presentation.

---

## Author Comment (AC1) · 22 May 2018

We acknowledge the constructive comments and suggestions made by Dr. González-Riancho.

Dr. Jorge León, a co-author of the study, has already answered to them and explained that they will be considered in the amended version of the article.

More specifically:

1) "The titles of the various sections in the method chapter would give a clearer idea of the work if they mention the method steps instead of generic terms valid for any

scientific study as "fieldwork" and "data analysis""

We have updated the names of the section following the suggestions made by Dr. González-Riancho, namely: Section 4: Material and Methods Section 4.1: Identification of micro-vulnerabilities along evacuation routes Section 4.2: Micro-vulnerabilities representation and classification

We hope these title would better describe the contents of each subsection.

2) "It is crucial that the authors clarify which parts from their work are original and which ones are not. The main differences between the two works carried out in Iquique should be clarified to better understand if there are scientific innovations in this work or if it is a case study applying the method from Leon and March (2016)"

Dr. Jorge León, has explained in his reply, the differences between the present contribution and the one previously published by León and March (2016). Indeed, in our article we propose a methodology intended to provide a quantitative assessment of urban micro-vulnerabilities; in their paper, León and March stated that there was a need to do so, but they do not attempt to develop a methodology.

We have added the following phrase in the second paragraph of section 4 (p. 6, lines 29-30) to clarify this, as suggested by the reviewer:

"Following the prior guidelines and as a complement to the more qualitative approach developed by León and March (2016)".

3) "In the Data analysis Section, two different classifications of the elements found in the evacuation routes are described. The first classification, based on (i) blockages, (ii) level changes, and (iii) surface roughness, seems a bit disconnected to the method described in pages 8-9. Only after reading the next section on friction rates (pages 10-11) the role of this classification is understood. It would be advisable to mention in page 8, lines 5-7, that this classification is used later for the calculation of the friction rates"

Dr. León has already replied to this comment, which we acknowledge. In order to facilitate the reading of the article, we have deleted the following phrase which appeared at the end of the first paragraph of subsection 4.2, and included an additional Figure (Figure 2 in the revised manuscript) which summarized the different steps of the methodology that is described in section 4.

Deleted: "Subsequently these elements were classified according to three principal criteria: i) blocking or decrease in spaces available for movement, ii) abrupt surface level changes, and iii) considerable changes in surface roughness."

The suggestions made by Dr. González-Riancho were well received and changes were made accordingly in the revised manuscript. We believe that the manuscript is now easier to read. We thank Dr. González-Riancho.

Please also note the supplement to this comment:
https://www.nat-hazards-earth-syst-sci-discuss.net/nhess-2017-458/nhess-2017-458-AC1-supplement.pdf

[Figure]

**1** ⎯⎯⎯⎯⎯⎯⎯⎯⎯⎯⎯⎯⎯⎯⎯⎯⎯⎯⎯⎯⎯

FIELDWORK
Detailed diagnosis of the current state of evacuation routes, video footage.

**2** ⎯⎯⎯⎯⎯⎯⎯⎯⎯⎯⎯⎯⎯⎯⎯⎯⎯⎯⎯⎯⎯

DATA ANALYSIS
Post-processing stage, micro-vulnerabilities mapping using GIS.

[Figure]

**3** ⎯⎯⎯⎯⎯⎯⎯⎯⎯⎯⎯⎯⎯⎯⎯⎯⎯⎯⎯⎯⎯

EVACUATION ROUTE OBSTRUCTION LEVEL
cuotient of the sum of all the areas of *micro-vulnerabilities* on a particular evacuation route, individually weighted by a speed reduction factor and the total surface area of the evacuation route.

$$\text{\textit{Friction Factor}} = \frac{\Sigma \text{ weighted \textit{micro-vulnerabilities} areas}}{\text{Total route area}}$$

**4** ⎯⎯⎯⎯⎯⎯⎯⎯⎯⎯⎯⎯⎯⎯⎯⎯⎯⎯⎯⎯⎯

DETERMINATION of the relative degree of vulnerability of evacuation routes.

[Figure]

**Fig. 1.** Added figure to summarize the methodology

**Supplement:**

[revised manuscript text omitted]

---

## Author Comment (AC2) · 22 May 2018

We acknowledge the suggestion made by Dr. Rahayu. We have prepared and additional figure to summarize the proposed methodology so that the reader could better appraise it in a graphical presentation as suggested by Dr. Rahayu. This figure is presented at the beginning of section 4.

We hope that this addition, along with other minor changes introduced to the manuscript also following suggestions made by Dr. González-Riancho, will be useful to improve the description of the methodology presented in the manuscript.

[Figure]

We thank the positive comments and suggestions made by Dr. Rahayu.

Please also note the supplement to this comment:
https://www.nat-hazards-earth-syst-sci-discuss.net/nhess-2017-458/nhess-2017-458-AC2-supplement.pdf

**①** FIELDWORK
Detailed diagnosis of the current state of evacuation routes, video footage.

**②** DATA ANALYSIS
Post-processing stage, micro-vulnerabilities mapping using GIS.

[Figure]

**③** EVACUATION ROUTE OBSTRUCTION LEVEL
cuotient of the sum of all the areas of *micro-vulnerabilities* on a particular evacuation route, individually weighted by a speed reduction factor and the total surface area of the evacuation route.

$$\frac{Friction}{Factor} = \frac{\Sigma \text{ weighted } micro\text{-}vulnerabilities \text{ areas}}{\text{Total route area}}$$

**④** DETERMINATION of the relative degree of vulnerability of evacuation routes.

[Figure]

**Fig. 1.** Graphical representation of the proposed methodology

**Supplement:**

[revised manuscript text omitted]

---

## Author Response (AR1)

Dr. Mauricio González
Editor
Natural Hazards and Earth System Science

Dear Dr. González

We acknowledge the positive comments and constructive suggestions made in the open discussion by the two reviewers (Dr. González-Riancho Calzada and Dr. Rahayu) to our article entitled "Identification and classification of urban micro-vulnerabilities in tsunami evacuation routes for the city of Iquique, Chile". We have considered them in the revised manuscript we are re-submitting for your consideration. We believe that these amendments should facilitate the reading of the paper and also clarify some issues that were identified by the reviewers.

Below, you will find detailed explanations about how the suggestions and criticisms raised by the reviewers were taken into account in the revised manuscript.

Similarly, we have also carried out a proof edition process to correct typos and improve the English along the manuscript. These changes, are highlighted in the "track changes" version of the updated manuscript.

Kind regards,

Dr. Rodrigo Cienfuegos
Associate Professor
Hydraulic and Environmental Department
School of Engineering
CIGIDEN Director
Pontificia Universidad Católica de Chile

I.       Answers comments from Dr. González-Riancho Calzada

**Specific Comments**

We thank Dr. González-Riancho Calzada for providing feedback for improving our manuscript. Here we provide answers to her specific comments and explain how we have amended the manuscript following her suggestions.

1. *"The method applied includes three steps: (i) diagnosis of evacuation routes, (ii) georeferencing and classification of micro-vulnerabilities, and (iii) calculation of the friction rate. The titles of the various sections in the method chapter would give a clearer idea of the work if they mention the method steps instead of generic terms valid for any scientific study as "fieldwork" and "data analysis"."*

R1:    We agree with this comment. The revised version of the manuscript includes more accurate titles for each section, more specifically:

Section 4: Methods > Material and Methods
Section 4.1: Fieldwork > Identification of micro-vulnerabilities along evacuation routes
Section 4.2: Data Analysis > Micro-vulnerabilities representation and classification

2. *"The work is based on a previous publication by Leon and March (2016) as mentioned in Page 6, and a great part of the method applied here follows that one. At least two of the three steps (diagnosis of the evacuation routes and classification of microvulnerabilities) seem to be based on criteria and categories defined in Leon and March (2016). This reviewer has not been able to have access to the work by Leon and March (2016), and consequently could not prove if the work presented by the authors is original or if it is a case study or a method already published. It is crucial that the authors clarify which parts from their work are original and which ones are not. The main differences between the two works carried out in Iquique should be clarified to better understand if there are scientific innovations in this work or if it is a case study applying the method from Leon and March (2016)."*

R2:    We agree that this work is referenced several times throughout the new manuscript (which may confuse some readers about the originality of the present contribution). However, we also point out that León and March's document introduce a more qualitative approach to the diagnosis of evacuation routes and classification of micro-vulnerabilities. Indeed, León and March (2016) underline the need for a more quantitative analysis for these issues, which is the starting point of our manuscript. We have provided in the online discussion a copy of the León and March (2016) paper so that the reviewer (and the editor) could evaluate the differences between both works and realize that the present contribution tackles new aspects that were not covered in León and March (2016).

To clarify this, we have added new sentences in section 4 in order to emphasis the latter, more specifically:

P6, L. 28-32: In the fieldwork carried out by León and March (2016), a series of vulnerable points on a micro-scale level were detected on evacuation routes in Iquique, which were classified (according to their origin) into three categories: i) precarious physical conditions and inadequate maintenance, ii) problems related to the design of the public space, and iii) inappropriate use of sidewalks, suggesting to conduct larger surveys focused in a quantitative analysis at this scale.

P7, L. 1-3: Following the prior guidelines and as a complement to the work of more qualitative approach developed by León and March (2016), we conduct here a detailed micro-scale analysis of the Iquique's urban context, as an attempt to characterize potential difficulties to carry out effective evacuation processes.

> 3. *"In the Data analysis Section, two different classifications of the elements found in the evacuation routes are described. The first classification, based on (i) blockages, (ii) level changes, and (iii) surface roughness, seems a bit disconnected to the method described in pages 8-9. Only after reading the next section on friction rates (pages 10-11) the role of this classification is understood. It would be advisable to mention in page 8, lines 5-7, that this classification is used later for the calculation of the friction rates."*

R3:     We agree with this comment. Content of page 9 is focused on introducing a classification scheme according to each micro-vulnerability's cause, not necessarily related to the characteristics of the micro-vulnerability itself (which in turn are the inputs for friction rates). In order to facilitate the reading of the manuscript, summarizing the different steps that are described in Section 4, and the logic behind them, we have included now a new figure (Figure 2 in P.7 of the revised manuscript, following suggestions by Dr. Rahayu).

Similarly, we have added the following sentence in P.8 L.14-15 following the suggestion made by the reviewer: These observations give rise to the micro-vulnerability classification that is described in the following subsections.

We have improved the description of the adopted classification in P. 11 L.6-7 as follows:
To this end, each micro-vulnerability was classified according to one of these categories: i) blocking or decrease in spaces available for evacuation, ii) abrupt surface level changes, and iii) noticeable changes in surface roughness.

**Technical comments**

> 1. *Page 2, line 9:  replace " ´ . . .at the coast Mori et al. (2013); Fraser et al. (2013)" with "at the coast (Mori et al., 2013; Fraser et al., 2013)"*

R1: Thank you, we have amended this in the revised manuscript.

> 2. *Page 4, line 19: please add ´ the year of the Mw 7.7 earthquake and tsunami in Mentawai Islands. 2010?*

R2: Thank you, we have added the date 2010. See the revised manuscript.

> 3. *Page 4, line 23: replace "spacial" with "special"*

R3: Thank you, we have amended this typo in the revised manuscript. The correct word is however, "spatial"

> 4. *Page 9, Table 1: Leon and March (2016) ´ should be cited in the table caption.*

R4: Thank you, we have added this reference in the revised manuscript.

**Additional amendments**

After revising the manuscript, we have corrected several other typos, and performed a careful proof reading of the English. The changes are highlighted in the "track changes" pdf version of the updated manuscript.

II.      Answers comments from Dr. Rahayu

1. *"In overall, the idea is very interesting and innovative in assessing and analyzing micro scale vulnerability which needed to mitigate in order to enhance designed capacity of evacuation planning. This will be very useful for developing countries with similar situation. The data and analysis is well presented, however the approach and methodology is not clear to achieve the objective of the paper. This will be good if presented in a clear graphical presentation."*

R1: We acknowledge the suggestion made by Dr. Rahayu. We have prepared and additional figure to summarize the proposed methodology so that the reader could better appraise it in a graphical presentation as suggested by Dr. Rahayu. This figure is presented at the beginning of section 4, in page 7.

We hope that this addition, along with other minor changes introduced in the manuscript also following suggestions made by Dr. González-Riancho Calzada, will be useful to improve the description of the methodology presented in the manuscript.

We thank the positive comments and suggestions made by Dr. Rahayu.

**Additional amendments**

After revising the manuscript, we have corrected several other typos, and performed a careful proof reading of the English. The changes are highlighted in the "track changes" pdf version of the updated manuscript.

[revised manuscript text omitted]